# Avoiding Inferior Clusterings with Misspecified Gaussian Mixture Models

## Abstract

In this paper, we examine the performance of both Expectation Maximization (EM) and Gradient Descent (GD) on unconstrained Gaussian Mixture Models when there is misspecification. Our simulation study reveals a previously unreported class of *inferior* clustering solutions, different from spurious solutions, that occurs due to asymmetry in the fitted component variances. We theoretically analyze this asymmetry and its relation to misspecification. To address the problem, we design a new functional penalty term for the likelihood based on the Kullback Leibler divergence between pairs of fitted components. Closed form expressions for the gradients of this penalized likelihood are difficult to derive but GD can be done effortlessly using Automatic Differentiation. The use of this penalty term leads to effective model selection and clustering with misspecified GMMs, as demonstrated through theoretical analysis and numerical experiments on synthetic and real datasets.

## 1 Introduction

The well-established paradigm of model-based clustering assumes data to be generated by a finite mixture model where each component represents a cluster. Gaussian Mixture Models (GMM), in particular, are widely used in a variety of applications (McLachlan & Peel, 2000). Expectation Maximization (EM) and its variants are, by far, the most popular methods to obtain Maximum Likelihood Estimates (MLE) of GMM parameters (Dempster et al., 1977; McLachlan & Krishnan, 2007).

The one-to-one correspondence between fitted components and clusters, that makes model-based clustering appealing, assumes that the underlying model is correctly specified and each data cluster can be viewed as a sample from a mixture component. In real data, the true distribution is rarely known and further, data may be contaminated by noise or outliers from a distribution different from the assumed model. In such *misspecified* settings, MLE may fail to recover the underlying cluster structure (Farcomeni & Greco, 2016).

The behaviour of EM for misspecified GMMs was recently studied by Dwivedi et al. (2018; 2020) who theoretically quantify the bias in estimates under univariate settings and specific cases, e.g., under- and over-specified number of components. They also characterize the convergence of EM iterates, which, for misspecified GMMs, converge to the Kullback-Leibler projection of the data-generating distribution onto the fitted model class, instead of approximating the true model parameters. Others have studied misspecifcation in the Bayesian setting, e.g., to find a modified likelihood that is robust to mild perturbations from the model (Miller & Dunson, 2018; Masegosa, 2020), to find the number of components (Miller & Harrison, 2018; Jitta & Klami, 2018) and to find identifiability conditions and consistency guarantees (Aragam et al., 2020). In this work, we study misspecified GMMs in a frequentist parametric setting, with the practical aim of improving clustering accuracy, i.e., inferring correct labels.

It is well known that MLE, even in the absence of misspecification, may give rise to spurious solutions that are local maximizers of the likelihood but without adequate interpretability of cluster structure (Ingrassia, 2004; García-Escudero et al., 2014). It is a consequence of the unboundedness of the GMM likelihood function for unrestricted component covariance matrices (Day, 1969), that results in solutions from EM with fitted 'degenerate' component(s) having very small variance corresponding to a cluster containing very few closely located data points; in the case of multivariate data, there are components with very small generalized

variance, lying in a lower-dimensional subspace and close to the boundary of the parameter space (Peel & McLachlan, 2000). Previous approaches to avoid spurious solutions include multiple restarts (Peel & McLachlan, 2000) and restricting the parameter space in EM to avoid degeneracy (Ingrassia, 2004; Ingrassia & Rocci, 2007; Chen & Tan, 2009; Ingrassia & Rocci, 2011).

Contamination through noise or outliers, which may be from a population different from the assumed model, may also lead to spurious solutions. Approaches to fit contaminated mixtures include trimming and restricting the parameter space (Cuesta-Albertos et al., 1997; Ruwet et al., 2013; García-Escudero et al., 2014). In addition, constrained GMMs have been proposed to tackle spurious solutions and contamination (Banfield & Raftery, 1993; Celeux & Govaert, 1995; Fraley & Raftery, 2002; Punzo & McNicholas, 2016); typical constraints enforce varying degrees of homoscedasticity across the component covariance matrices in terms of volume, orientation and/or shape. Such constrained models reduce the number of parameters to be estimated, and thereby enable efficient implementations, e.g., in Mclust (Scrucca et al., 2016). However, imposing such hard modeling constraints may be too stringent and can degrade clustering performance as they do not take into account feature dependencies that can vary across the clusters, and may lead to incorrect orientations or shapes of the inferred clusters. As a result, pre-determined constraints on means or covariance matrices are discouraged and a more data-dependent discovery of cluster structure is recommended, e.g., by Zhou et al. (2009); Fop et al. (2019).

In a different line of work, that, to our knowledge, has not studied misspecification, gradient-based methods for MLE of GMMs have been investigated (Redner & Walker, 1984; Boldea & Magnus, 2009; Montanari & Viroli, 2011; Alexandrovich, 2014). In general, EM has numerous advantages over gradient-based methods as discussed in Xu & Jordan (1996). More recently, there has been considerable development of Automatic Differentiation for Gradient Descent (AD-GD), that obviate the need to derive closed-form expressions of gradients and thereby facilitate inference in complex models such as deep neural networks as well as mixture models (Maclaurin et al., 2015; Kasa & Rajan, 2022).

In our study, we first compare the clustering performance – in terms of Adjusted Rand Index (ARI) (Hubert & Arabie, 1985) – of GD and EM with misspecified GMMs. Our simulations reveal a previously unreported class of poor clustering solutions, with both EM and GD. These solutions, that we call *inferior*, occur frequently in MLE procedures (irrespective of the initialization), have asymmetric fitted covariances that vary in their sizes and have poor cluster interpretability. Thus, they differ in their characteristics from spurious solutions. More details are given in section 3. Our theoretical analysis on a specific setting of univariate mixtures and under-specified number of components yields evidence on the connection between asymmetry of fitted components and misspecification. It also motivates the design of a new penalty term, based on the Kullback Leibler divergence between pairs of fitted component Gaussians, to avoid inferior clusterings. Closed forms for the gradients of this penalized likelihood are difficult to derive and we leverage AD-GD to develop algorithms[1] for clustering (SIA) and a model selection criterion (MPKL). Our functional regularization approach, that enforces soft constraints tunable by varying the hyperparameters, avoids both spurious solutions and shortcomings of imposing hard modeling constraints. Our extensive experiments demonstrate the advantage of our methods in clustering with misspecified GMMs.

To summarize, our contributions in this paper are:

- We conduct an empirical analysis to compare the clustering performance of GD and EM inference on unconstrained and misspecified GMMs.

- We identify and characterize the problem of *inferior* clusterings that have high likelihood and low ARI, similar to spurious solutions, in both EM and GD. Unlike spurious solutions, these solutions have fitted components that are markedly asymmetric with respect to their orientation and sizes, and occur frequently with many different initializations. We theoretically examine how asymmetry of fitted components varies with misspecification.

- We propose a new penalty term that is designed to avoid inferior solutions and prove that the penalized likelihood is bounded and hence avoids degeneracy. Using this penalty term, we develop MPKL, a model selection criterion and SIA, an AD-GD based clustering algorithm.

---

[1]Python implementation given in Supplementary Material

- Experiments on synthetic and real datasets demonstrate the advantages of SIA, over previous clustering methods with both constrained and unconstrained GMMs, in cases of misspecification.

## 2 Background

Let $f(\boldsymbol{x}; \boldsymbol{\theta})$ be the density of a $K$-component mixture model. Let $f_k$ denote the $k^{\text{th}}$ component density with parameters $\boldsymbol{\theta}_k$ and weight $\pi_k$. The density of the mixture model is given by $f(\boldsymbol{x}; \boldsymbol{\theta}) = \sum_{k=1}^{K} \pi_k f_k(\boldsymbol{x}; \boldsymbol{\theta}_k)$, where $\sum_{k=1}^{K} \pi_k = 1$ and $\pi_k \geq 0$ for $k = 1, \dots, K$ and $\boldsymbol{\theta}$ denotes the complete set of parameters. In a GMM, each individual component $f_k$ is modeled using a multivariate Gaussian distribution $\mathcal{N}(\boldsymbol{\mu}_k, \boldsymbol{\Sigma}_k)$ where $\boldsymbol{\mu}_k$ and $\boldsymbol{\Sigma}_k$ are its mean and covariance respectively. Appendix A lists all the symbols used herein.

Given $n$ independent and identically distributed (*iid*) instances of $p$-dimensional data, $[x_{ij}]_{n \times p}$ where index $i$ is used for observation, and $j$ is used for dimension, Maximum Likelihood Estimation (MLE) aims to find parameter estimates $\hat{\boldsymbol{\theta}}$ from the overall parameter space $\boldsymbol{\Theta}$ of $f(\boldsymbol{\theta})$ such that probability of observing the data samples $\mathbf{x}_1, \dots, \mathbf{x}_n$ is maximized, i.e., $\hat{\boldsymbol{\theta}} = \arg\max_{\boldsymbol{\theta} \in \boldsymbol{\Theta}} \mathcal{L}(\boldsymbol{\theta})$, where, $\mathcal{L}(\boldsymbol{\theta}) = \frac{1}{n} \sum_i \log f(\mathbf{x}_i; \boldsymbol{\theta})$ is the empirical expected loglikelihood.

Following White (1982), if the observed data are $n$ *iid* samples from a probability distribution $P(\eta^*)$ (where $\eta^*$ denotes the *true* set of parameters) and the fitted model has the same functional form $P(.)$, then the model is said to be correctly specified. Otherwise, the model is said to be *misspecified*. Note that when the number of dimensions are greater than the number of datapoints ($p > n$), there are not enough datapoints to determine if the fitted model and data-generating distribution are parameterized by the same model, even along a single dimension and so, the notion of misspecification is moot. Appendices B and C give an overview of, respectively, spurious solutions and likelihood-based model selection criteria (such as Akaike Information Criterion (AIC) and Bayesian Information Criterion (BIC)).

Recent reviews on Automatic Differentiation (AD) can be found in Baydin et al. (2018); Margossian (2019); a brief review is in Appendix D. To obtain MLE of GMMs, EM elegantly solves 3 problems: (1) Intractability of evaluating the closed-forms of the derivatives, (2) Ensuring positive definiteness of the covariance estimates $\hat{\boldsymbol{\Sigma}}_k$, and (3) Ensuring the constraint on the component weights ($\sum_k \hat{\pi}_k = 1$). *Problem 1* is inherently solved by the use of AD. *Problems 2* and *3* can be addressed through simple reparametrizations in the form of matrix decomposition and the log-sum-exp trick respectively (Salakhutdinov et al., 2003; Kasa & Rajan, 2022). Details are in Appendix D. Constraints used in EM to mitigate spurious solutions and contamination can also be used in AD-GD through changes in the update equations.

## 3 Inferior Clustering Solutions

In this section, we illustrate inferior clustering solutions through an empirical comparison of EM and AD-GD in cases of misspecification. We also theoretically analyze the role of asymmetric fitted components in misspecified Gaussian mixture models.

### 3.1 Simulation Study

We study the clustering solutions obtained by fitting misspecified GMM on Pinwheel data, also known as warped GMM, with 3 components and 2 dimensions (Iwata et al., 2013). Pinwheel data is generated by sampling from Gaussian distributions, and then stretching and rotating the data. The centers are equidistant around the unit circle. The variance is controlled by parameters $r$, the radial standard deviation and $t$, the tangential standard deviation. The warping is controlled by a third parameter, $s$, the rate parameter. Thus, the extent of misspecification (i.e., the deviation of the data from the assumed Gaussian distributions in GMM) can be controlled using these parameters. An example is shown in Fig. 1. We generate 1800 Pinwheel datasets with different combinations of parameters. In addition, we also simulate 1800 3-component, 2-dimensional datasets from GMM with varying overlap of components (to analyze as a control case where there is no misspecification in fitting a GMM). For each dataset we obtain two clustering solutions, one each using EM and AD-GD. We run both the algorithms till convergence, using the same initialization

and stopping criterion. We use ARI to evaluate performance where higher values indicate better clustering. Details are in Appendix E.

Our experiments on these 3600 datasets show that EM outperforms AD-GD in both cases – when there is misspecification and no missspecification. However, when there is misspecification, we find that for both EM and AD-GD, there are many *inferior* solutions that have low ARI and unequal fitted covariances, often with one fitted component having relatively large covariance, resulting in a high degree of overlap between components. We illustrate this using Pinwheel data generated with parameters $r = 0.3$, $t = 0.05$ and $s = 0.4$. Both EM and AD-GD are run with 100 different initializations. We group the solutions into 4 sets based on AIC as shown in Table 1, which also shows the average AIC and ARI obtained by AD-GD and the number of EM and AD-GD solutions in each set. Table 27 (in Appendix E) show the statistics for EM solutions which are very similar. We visualize the clustering for one solution from each of these sets in Fig. 1.

Table 1: Summary statistics of clustering solutions over 100 random initializations on the Pinwheel dataset (see Fig. 1), grouped into 4 sets based on AIC ranges. Mean (Standard Deviation) of parameter estimates from AD-GD, EM estimates (in Appendix E) are similar.

| Set | AIC Range | AD-GD | | | | | | | | | # Solutions | |
|---|---|---|---|---|---|---|---|---|---|---|---|---|
| | | AIC | ARI | $\pi_1$ | $\pi_2$ | $\pi_3$ | $\|\mathbf{\Sigma}_1\|$ | $\|\mathbf{\Sigma}_2\|$ | $\|\mathbf{\Sigma}_3\|$ | | EM | AD-GD |
| 1 | 771-773 | 771.9 | 0.625 | 0.257 | 0.265 | 0.477 | 0.0002 | 0.0005 | 0.123 | | 24 | 19 |
| | | (6e-8) | (2e-16) | (3e-6) | (3e-6) | (1e-6) | (1e-9) | (4e-9) | (4e-7) | | | |
| 2 | 781-782 | 781.1 | 0.912 | 0.306 | 0.341 | 0.352 | 7e-4 | 0.01 | 0.01 | | 4 | 3 |
| | | (3e-6) | (0) | (1e-5) | (3e-5) | (1e-5) | (5e-9) | (5e-9) | (5e-7) | | | |
| 3 | 786-787 | 786.8 | 0.652 | 0.257 | 0.267 | 0.475 | 2e-4 | 5e-4 | 0.156 | | 16 | 27 |
| | | (2e-7) | (0) | (5e-6) | (3e-6) | (3e-6) | (3e-9) | (3e-9) | (2e-6) | | | |
| 4 | 788-850 | 815.0 | 0.806 | 0.28 | 0.315 | 0.403 | 6e-4 | 3e-3 | 0.047 | | 56 | 51 |
| | | (17.84) | (0.06) | (0.015) | (0.010) | (0.024) | (4e-4) | (1e-3) | (0.018) | | | |

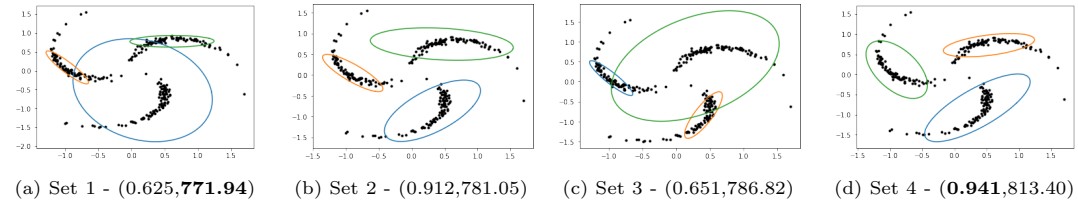

(a) Set 1 - (0.625,**771.94**)    (b) Set 2 - (0.912,781.05)    (c) Set 3 - (0.651,786.82)    (d) Set 4 - (**0.941**,813.40)

Figure 1: 4 clustering solutions obtained with AD-GD using different initializations on Pinwheel data; Sets refer to groups in Table 1; in parentheses: (ARI, AIC), best values in bold.

We observe that both EM and AD-GD obtain very similar solutions in terms of AIC and ARI for this dataset. The best *average* ARI is that of set 2 (row 2 of Table 1). These solutions are obtained in less than 5% of the cases. The *overall best* ARI, of 0.941, is from a solution in set 4 that has a high AIC value of 813.4 as shown in Fig. 1 (d). The best AIC, of 771.9, is obtained by a solution from set 1 which has considerably lower ARI of 0.625 as shown in Fig. 1 (a). Thus, we see solutions with high likelihood and low ARI. We observe that there are many such inferior solutions in sets 1, 3 and 4 having a fitted component with large variance, also seen in the specific solutions in Fig. 1.

In cases of misspecification, inferior clusterings from both EM and AD-GD, are found to have fitted covariances that differ considerably in their orientations and sizes. We observe this in the solutions in Fig. 1 and through the summary statistics in Table 1. We find that such inferior solutions in misspecified models occur frequently with many different initializations, and typically when there is a component with large variance. This is different from the characterization of spurious solutions (see Appendix B) that are found to occur rarely, only with certain initializations and due to a component with small variance (McLachlan & Peel, 2000).

We find similar inferior solutions with low ARI and low AIC in cases where Gaussian components are contaminated by a Student's-t distribution and random noise (details in Appendix F). Further, we have observed similar effects of misspecification in higher dimensions as well. Illustrations in datasets of up to 100 dimensions are in Appendix F.

### 3.2 Misspecification and Asymmetric Components

We now theoretically examine how asymmetry of fitted components varies with misspecification in the case of univariate mixtures, following the setting of Balakrishnan et al. (2017); Dwivedi et al. (2018; 2020).

Let the true data generating distribution be $G^* = \pi\mathcal{N}(-\mu, \sigma^2) + \pi\mathcal{N}(\mu, \sigma^2) + (1 - 2\pi)\mathcal{N}(\mu, b^2\sigma^2)$, with $0 < \pi \leq 0.5$. Without loss of generality, we assume $\mu > 0$. When $\pi = 0.5$, the true distribution is a symmetric 2-component GMM, with the component means equidistant on either side of the real line as shown in Fig. 2a. In this case, it has been shown that under some identifiability conditions, fitting a 2-component GMM (i.e., without any misspecification) on the data sampled from the true distribution using MLE leads to symmetric fitted components whose parameters converge to that of the true distribution White (1982); Xu & Jordan (1996). As $\pi$ is reduced from 0.5, an additional Gaussian component with mean $\mu$ (that coincides with one of the component means) and variance $b^2\sigma^2$ is introduced. Fitting a 2-component GMM in the case when $\pi < 0.5$ leads to a misspecified model. We analyze the asymmetry of the fitted components for $\pi < 0.5$ with varying misspecification ($b$).

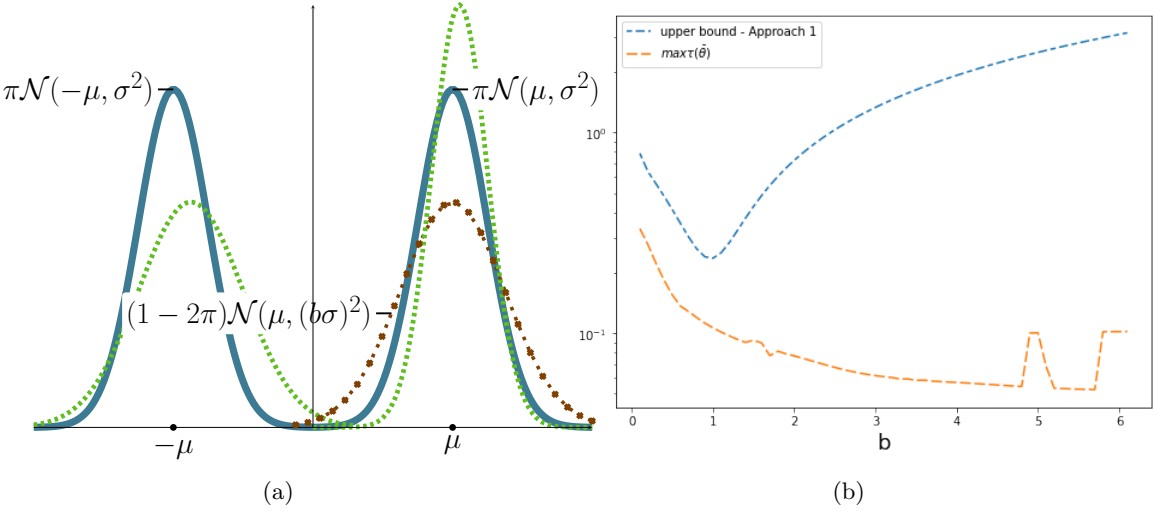

(a)                                          (b)

Figure 2: a) True distribution is a GMM ▬ and a contamination ✖. We fit a 2-component (misspecified) GMM - - -. b) Empirical comparison of $\tau(\bar{\boldsymbol{\theta}})$ and its upper bound with varying $b$.

Let the fitted misspecified distribution be $G'(\bar{\boldsymbol{\theta}}) = (1 - \bar{\pi})\mathcal{N}(\bar{\mu}_1, \bar{\sigma}_1^2) + \bar{\pi}\mathcal{N}(\bar{\mu}_2, \bar{\sigma}_2^2)$, where $\bar{\boldsymbol{\theta}} = (\bar{\mu}_1, \bar{\sigma}_1^2, \bar{\mu}_2, \bar{\sigma}_2^2, \bar{\pi})$ and $\bar{\boldsymbol{\theta}} \in \arg\min_{\theta \in \Theta} \mathrm{KL}\left(G_*, G'(\boldsymbol{\theta})\right)$. Note that this is a projection to the fitted model and is the best possible estimator (Dwivedi et al., 2018). Let erf be the Gauss error function (Chang et al., 2011) which is an odd function, whose values lies in $(-1, 1)$ and is related to the CDF of a standard normal distribution as $\Phi(x) = \frac{1}{2}\left[1 + \mathrm{erf}\left(\frac{x}{\sqrt{2}}\right)\right]$.

When misspecification is small, the means of fitted components $\bar{\mu}_1, \bar{\mu}_2$ typically have opposite signs, as the true components flank the y-axis. When there is misspecification, we expect $\mathrm{erf}\left(\frac{\bar{\mu}_1}{\sqrt{2}\bar{\sigma}_1}\right)$ and $\mathrm{erf}\left(\frac{\bar{\mu}_2}{\sqrt{2}\bar{\sigma}_2}\right)$ to have unequal and opposite signs. We define an asymmetry coefficient:

$$\tau(\bar{\boldsymbol{\theta}}) = (\bar{\pi})\,\mathrm{erf}(\frac{\bar{\mu}_2}{\sqrt{2}\bar{\sigma}_2}) + (1 - \bar{\pi})\,\mathrm{erf}(\frac{\bar{\mu}_1}{\sqrt{2}\bar{\sigma}_1}).$$

$\tau(\bar{\boldsymbol{\theta}})$ measures a form of asymmetry in $\frac{\mu}{\sigma}$ of the fitted components when there is misspecification. When there is no misspecification, since MLE estimates converge to the true parameters asymptotically, $\tau(\bar{\boldsymbol{\theta}})$ also converges to zero. However, when there is misspecification, the variance of one of the components, say $\sigma_1$, tends to be much larger than that of the other component (as seen in table 1); in such cases, $\mathrm{erf}\left(\frac{\mu_1}{\sqrt{2}\sigma_1}\right)$ would be much closer to zero compared to the error function for the other component and $\tau(\bar{\boldsymbol{\theta}})$ would converge to a non-zero value. We derive bounds on $\tau(\bar{\boldsymbol{\theta}})$ as follows (proof in Appendix G):

**Theorem 1.** *Let the true data generating distribution be* $G^* = \pi \mathcal{N}(-\mu, \sigma^2) + \pi \mathcal{N}(\mu, \sigma^2) + (1 - 2\pi)\mathcal{N}(\mu, b^2\sigma^2)$, *with* $0 < \pi \leq 0.5$. *Let the fitted misspecified distribution be* $G'(\bar{\theta}) = (1 - \bar{\pi})\mathcal{N}(\bar{\mu}_1, \bar{\sigma}_1^2) + \bar{\pi}\mathcal{N}(\bar{\mu}_2, \bar{\sigma}_2^2)$, *with asymmetry coefficient* $\tau(\bar{\theta})$, *where* $\bar{\theta} = (\bar{\mu}_1, \bar{\sigma}_1^2, \bar{\mu}_2, \bar{\sigma}_2^2, \bar{\pi})$ *and* $\bar{\theta} \in \arg\min_{\theta \in \Theta} \text{KL}(G^*, G'(\theta))$. *Let* $C_2 := (1 - 2\pi)(-\log b + 0.5b^2 - 0.5) + \pi \frac{2\mu^2}{\sigma^2}$ *and* $C_w := \frac{1-2w}{2} erf\left(\frac{-\mu}{\sqrt{2}\sigma}\right)$. *Then,*

$$-\sqrt{2C_2} + 2C_w \leq \tau(\bar{\theta}) \leq \sqrt{2C_2} + 2C_w.$$

Note that $C_2$ and $C_w$ depend only on the true parameters $(\pi, \mu, \sigma, b)$ and are constants with respect to the fitted model. Assuming that the true parameters are known, these bounds provide a certification on whether the fitted parameters $\bar{\theta}$ indeed correspond to the maximum likelihood, thus helping to filter out fitted parameters corresponding to spurious solutions and undesirable local optima.

We empirically compare the upper bound and observed value of $\tau(\bar{\theta})$ when $\mu = 5.0, \sigma = 10, \pi = 0.35$ by varying the values of $b$ (the behaviour of the lower bound is qualitatively similar). For a given value of $b \in \{0.1, 0.2, \ldots, 6\}$, we simulate 50 datasets and pick the maximum of $\tau(\bar{\theta})$ among these 50 output solutions. The upper bound and the observed maximum are plotted in Fig. 2b. At around $b = 1$ (when there is no misspecification), the upper bound reaches its minimum. As $b$ moves away from 1, the upper bound increases which illustrates that as misspecification increases, the asymmetry in the fitted components also increases. Additional plots for other values of $(\mu, \sigma, \pi)$ are in Appendix G. We observe that, if one of the components has a relatively large variance, then many datapoints may be wrongly assigned to this component leading to poor clustering.

## 4 A Penalized Clustering Method

### 4.1 A New Penalty Term

Our analysis in the previous section naturally leads to the goal of reducing the asymmetry in the fitted components to improve clustering with misspecified GMMs. To this end, we develop a new functional penalty term that (i) penalizes differences in orientations and sizes of the fitted components to avoid inferior solutions and (ii) bounds the penalized likelihood to avoid spurious solutions during ML estimation.

Our penalty term is based on the KL-divergence between component Gaussians. Let $\mathcal{N}_k$ denote the multivariate Gaussian distribution $\mathcal{N}(\boldsymbol{\mu}_k, \boldsymbol{\Sigma}_k)$. The KL-divergence $KL(\mathcal{N}_1, \mathcal{N}_2)$ is given below where each term can provide useful penalties:

$$\frac{1}{2}\big[\underbrace{\log \frac{|\boldsymbol{\Sigma}_2|}{|\boldsymbol{\Sigma}_1|}}_{A} + \underbrace{\text{tr}\{\boldsymbol{\Sigma}_2^{-1}\boldsymbol{\Sigma}_1\} - p}_{B} + \underbrace{(\boldsymbol{\mu}_2 - \boldsymbol{\mu}_1)^T \boldsymbol{\Sigma}_2^{-1}(\boldsymbol{\mu}_2 - \boldsymbol{\mu}_1)}_{C}\big]$$

- A: Penalizes the difference in size of the covariance matrices of the two components. Even if the directions of principal axes of the two covariance matrices are different, this term will tend to zero if the component determinants are similar, since $\log(1) = 0$.

- B: Penalizes the difference in orientations, i.e., if the directions of principal axes of the component covariance matrices are vastly different. When $\boldsymbol{\Sigma}_1$ and $\boldsymbol{\Sigma}_2$ are equal, this penalty term becomes zero.

- C: Penalizes the assignment of a single cluster to faraway outlier points which have an extremely low likelihood of being observed. If some outlier points are assigned a single cluster, then the cluster center, $\mu_1$, would be different from the cluster center $\mu_2$ of non-outlier data, and so, the Mahalonobis distance between the cluster centers, $\|\mu_1 - \mu_2\|$, would be high, with higher penalization.

KL-divergence is not symmetric about $\boldsymbol{\Sigma}_1$ and $\boldsymbol{\Sigma}_2$. If there is an order of magnitude difference in the covariance matrices $\boldsymbol{\Sigma}_1$ and $\boldsymbol{\Sigma}_2$, it can be detected through the values of $KL(\mathcal{N}_1, \mathcal{N}_2)$ and $KL(\mathcal{N}_2, \mathcal{N}_1)$. The difference in their values primarily stems from the difference between the terms $(\boldsymbol{\mu}_2 - \boldsymbol{\mu}_1)^T \boldsymbol{\Sigma}_2^{-1}(\boldsymbol{\mu}_2 - \boldsymbol{\mu}_1)$ and $(\boldsymbol{\mu}_2 - \boldsymbol{\mu}_1)^T \boldsymbol{\Sigma}_1^{-1}(\boldsymbol{\mu}_2 - \boldsymbol{\mu}_1)$, and $\text{tr}\{\boldsymbol{\Sigma}_2^{-1}\boldsymbol{\Sigma}_1\}$ and $\text{tr}\{\boldsymbol{\Sigma}_1^{-1}\boldsymbol{\Sigma}_2\}$. The difference in these two KL divergence values provides signal about the overlap or asymmetry of the two Gaussians. This notion is generalized to

a $K$-component GMM, through the combinatorial KL divergences, KLF (forward) and KLB (backward), where $1 \leq k_1, k_2 \leq K$: $KLF := \sum_{k_1 < k_2} KL(\mathcal{N}_{k_1}, \mathcal{N}_{k_2})$; $KLB := \sum_{k_2 < k_1} KL(\mathcal{N}_{k_1}, \mathcal{N}_{k_2})$. Well separated clusters typically have equal and similar values of KLF and KLB. We denote both the values by $KLDivs = \{KLF, KLB\}$. We note that these two sums (KLF + KLB) together give the sum of Jeffrey's divergence between all components Budka et al. (2011). In the clustering outputs shown in Fig. 1, we see that in solution (c) from set 3, where clustering is poor, KLF= 258 and KLB= 494 (the difference is high), while in solution (d) from set 4, which has better clustering, KLF= 127 and KLB= 64 (with low difference).

Our proposed penalty term is a weighted sum of the $KLF$ and $KLB$ terms: $-w_1 \times KLF - w_2 \times KLB$, with negative weights $(-w_1, -w_2)$. With positive weights GD will further shrink the smaller clusters. Negative weights lead to reduced overlap as well as clusters of similar volume. In our experiments, we found that choosing $w_1 = w_2$ works in almost all cases. Further, choosing $w_1 = w_2$ makes the penalized objective invariant to permutation in the component labels.

Optimization of likelihood with these penalty terms can be implemented effortlessly through AD-GD where gradients in closed forms are not required. However, the use of such complex penalties is difficult within EM. We cannot obtain closed forms of the covariance estimates. Closed forms for the mean update can be derived (Appendix H) but is laborious and each mean update depends on means for all other components, and hence cannot be parallelized in EM.

## 4.2 Sequential Initialization Algorithm (SIA)

SIA consists of two steps. In Step I of SIA, we use the loglikelihood $\mathcal{L}$ as the objective and run EM or AD-GD to fit a GMM. Typically, the output at the end of Step I will have unequal KL-divergences for misspecified models. The parameters at the end of Step I are used to initialize the algorithm in step II where we modify the objective function to:

$$\mathcal{M} = \mathcal{L} - w_1 \times KLF - w_2 \times KLB \tag{1}$$

After the second optimization step the likelihood decreases but the KL-divergence values, KLF and KLB, come closer. The complete algorithm is presented in Algorithm 1.

---

**Algorithm 1** SIA

**Input:** Data: $n \times p$ dimensional matrix, number of clusters $K$, learning rate $\epsilon$, convergence tolerance $\gamma$.
**Initialize** at $t = 0$: $\hat{\boldsymbol{\mu}}_k^0, \hat{\alpha}_k$'s using K-Means or random initialization; $\hat{\mathbf{U}}_k^0$ (for GMM) as identity matrices.
**Step I:** Run either AD-GD or EM with Likelihood $\mathcal{L}$ as objective
**Step II:**
Set $\mathcal{M}$ (Eq. 1) as the objective
**Initialize:** Use Output of Step I
**REPEAT:** At every iteration $t + 1$:

$$\hat{\alpha}_k^{t+1} := \hat{\alpha}_k^t + \epsilon \frac{\partial \mathcal{M}}{\partial \alpha_k}; \ \hat{\pi}_k^{t+1} := \frac{e^{\hat{\alpha}_k^{t+1}}}{\sum_{k'} e^{\hat{\alpha}_{k'}^{t+1}}}; \ \hat{\boldsymbol{\mu}}_k^{t+1} := \hat{\boldsymbol{\mu}}_k^{t+1} + \epsilon \frac{\partial \mathcal{M}}{\partial \boldsymbol{\mu}_k};$$

$$\hat{\mathbf{U}}_k^{t+1} := \hat{\mathbf{U}}_k^t + \epsilon \frac{\partial \mathcal{M}}{\partial \mathbf{U}_k}; \ \hat{\boldsymbol{\Sigma}}_k^{t+1} := \hat{\mathbf{U}}_k^{t+1} \hat{\mathbf{U}}_k^{t+1^T}$$

**UNTIL:** convergence criterion $|\mathcal{M}^{t+1} - \mathcal{M}^t| < \gamma$ is met

---

In SIA, after Step I we know the likelihood and model parameters of the unpenalized fitted model. In Step II, we choose $w_1, w_2$ in such a way that the overall likelihood of the model doesn't reduce drastically, to find solutions (with less dissimilar covariances) close to the solution after Step I. In our experiments, the values of $w_1, w_2$ have been kept equal and chosen from $\{0, 0.25, 0.5, 1, 1.25\}$ using the MPKL criterion (described below).

### 4.3 SIA Objective is Bounded

We consider maximum likelihood estimation of parameters $\boldsymbol{\theta}$ of a two-component GMM, from $n$ samples, each of $p$-dimensions, using SIA. We prove that our penalized log-likelihood $\mathcal{M}$ is bounded, and thus SIA alleviates the problem of degeneracy and spurious solutions in unconstrained GMMs.

**Theorem 2.** *Let $C$ denote a constant dependent only on $p, n, w_1, w_2, c$ and independent of $\boldsymbol{\theta}$. Assume that the spectral norm of the generalized variances obtained using MLE, $\boldsymbol{\Sigma}_1, \boldsymbol{\Sigma}_2$, is bounded by $c < \infty$ (as a regularity condition) and, without loss of generality, assume $|\boldsymbol{\Sigma}_1| \le |\boldsymbol{\Sigma}_2|$ and that the loglikelihood $\mathcal{L} \to \infty$ when the first component collapses on a datapoint, i.e., $|\boldsymbol{\Sigma}_1| \to 0$. For non-negative weights $w_1, w_2$, and any iteration $t$, SIA objective function $\mathcal{M}$ is always bounded:*

$$\mathcal{M} \le -\frac{n}{2}(1 + w_2)\left(\log(|\frac{w_2}{1+w_2}\boldsymbol{\Sigma}_2|) + p\right) + C.$$

The proof, in Appendix G, also clarifies that the result generalizes to arbitrary number of components. The bound is given in terms of $w_2$ because we assume $|\boldsymbol{\Sigma}_1| \le |\boldsymbol{\Sigma}_2|$ (the objective is not symmetric with respect to $w_1$ and $w_2$). Hence, $w_1$ does not appear in the theorem statement. Also note that, $w_1$ and $w_2$ are the only controllable hyper-parameters. By setting $w_2 = 0$, we can see that the maximum of unpenalized likelihood is unbounded. If a $\boldsymbol{\Sigma}$ collapses, the log-likelihood may increase but the trace term and the determinant in the penalization will explode. Hence, by having negative weights $-w_1, -w_2$ we can ensure that $\mathcal{M}$ is bounded and control the behavior of the algorithm. Note that both $|\boldsymbol{\Sigma}_1|, |\boldsymbol{\Sigma}_2|$ will not collapse to zero in a maximum-likelihood estimation approach because if both collapse the log-likelihood goes to negative infinity (which is easy to verify). A visualization of the likelihood surface, with and without penalty terms, is shown in Appendix I. The fact that there *exists* an upper bound to the penalized likelihood (even when the generalized variance of one of the components collapses to zero), irrespective of its tightness, is sufficient to show that SIA avoids degenerate and spurious solutions.

### 4.4 MPKL: A Model Selection Criterion

We define the **M**aximum absolute **P**airwise difference between **KL** divergence values (MPKL) for a $K$-component GMM as:

$$\max_{1 \le k_1, k_2 \le K} |KL\left(\mathcal{N}_{k_1}, \mathcal{N}_{k_2}\right) - KL\left(\mathcal{N}_{k_2}, \mathcal{N}_{k_1}\right)| \tag{2}$$

MPKL is an indicator of how well the clusters are separated. It is invariant to permutation in cluster labels and can be used as a criterion for selecting the number of clusters. For a chosen range of number of clusters $(2, \dots, L)$, we compute MPKL for each value and choose $K$ that minimizes MPKL: $\text{argmin}_{K \in [2, \dots, L]} MPKL$. Note that MPKL criterion is independent of SIA and can be potentially useful for any GMM-based method, even in the absence of suspected misspecification. In our experiments (§6 and Appendix L.2), we show that use of MPKL aids both GD and EM based methods.

### 4.5 Computational Complexity

The computational complexity is dominated by evaluating $KLF$ and $KLB$ which involves $O(K^2)$ matrix inversion steps ($O(p^3)$). Therefore, the overall time complexity of both SIA and MPKL is $O(K^2 p^3)$. This is comparable to most GMM inference methods (e.g., EM) due to the $O(p^3)$ matrix inversion step. Wall-clock times of EM, AD-GD and SIA implementations are compared in Appendix J.

### 4.6 Illustrative Examples

Fig. 3 shows the clustering solution (set 3) obtained on the Pinwheel data discussed in §3, after steps I and II of SIA. After step II, compared to the clustering after step I, the likelihood decreases, both the KLF and KLB values decrease, the ARI increases, and the clusters have less overlap. The same is observed for the other three sets of clustering solutions. Similar illustrations on datasets contaminated with Student's-t and random noise are in Appendix F.

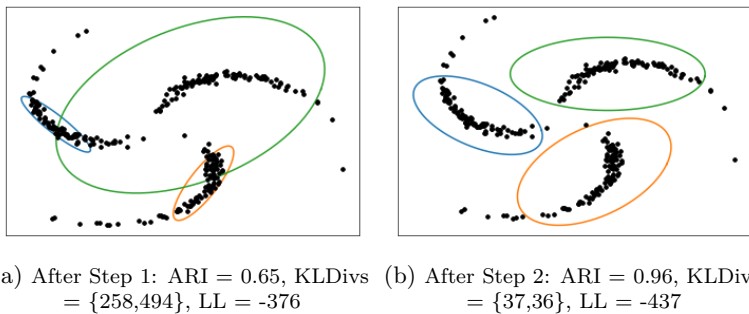

(a) After Step 1: ARI = 0.65, KLDivs
= {258,494}, LL = -376

(b) After Step 2: ARI = 0.96, KLDivs
= {37,36}, LL = -437

Figure 3: Clustering using SIA: compare with Fig. 1.

## 5  Simulation Studies

We simulate over 2000 datasets from mixtures with non-Gaussian components and with varying (i) cluster separation (ii) covariance structures and imbalance across component sizes and (iii - iv) number of components. In all these settings, we vary the dimensionality as well. Table 2 shows a summary of all the settings and the sections (§) in which the experiments are discussed. Experiments to evaluate MPKL as a model selection criterion are in Appendix L.2.

Table 2: Summary of Simulation Experiments; $K$: no. of components, $p$: dimensionality, $n$: no. of data points, $r$: ratio of weights, $\mu$: means, $\Sigma$: covariances. In each case we use $p = 2 \ldots 20$.

|   | Condition(s) varied | $K$ | $\mu$ | $\Sigma$ | $r$ | $n$ | § |
|---|---|---|---|---|---|---|---|
| 1 | Cluster Separation | 3 | varied | unit | 1 | $300p$ | 5.1 |
| 2 | Covariance and mixture weights | 2 | fixed | varied | {0.2, 0.5, 1} | {120p, 150p, 200p} | 5.2 |
| 3 | #components | under-specified | fixed | unit | 1 | 150 | L.1.1 |
| 4 | #components | over-specified | fixed | unit | 1 | 100 | L.1.2 |

We evaluate the clustering performance of misspecified GMM using four inference techniques – EM , AD-GD, MClust (that implements EM for GMMs with 14 different, including equi-covariance, constraints) and our SIA algorithm. Note that in SIA, EM and AD-GD, the same model (unconstrained GMM) is fitted; the difference lies only in the inference approach. MClust fits different models (constrained GMMs) altogether. We use it as a baseline method to compare the performance of the softer regularization-based approach in SIA with the harder constraint-based approach in MClust, two different ways of providing inductive bias to the models.

All the algorithms are initialized using K-Means and are run till convergence. In MClust, instead of the default initialization with model-based agglomerative clustering, we use K-Means for a fair comparison; the best among its 14 internal models are selected using BIC.

### 5.1  Varying Cluster Separation

We simulate data from p-dimensional mixture model with 3 Gaussian components, $\mathcal{N}(0.5\lambda\,(1,\ldots,1)_p, \mathbb{I}_p)$, $\mathcal{N}((0,\ldots,0)_p, \mathbb{I}_p)$, $\mathcal{N}(-0.5\lambda(1,\ldots,1)_p, \mathbb{I}_p)$, where $\lambda$ is a scaling factor which controls the cluster separation and $\mathbb{I}_2$ is a unit covariance matrix. We evaluate the performance for $\lambda$ values in $\{3, 4, 5, 7\}$ and $p$ values in $\{2, 3, 5, 10, 20\}$. For each value of $p$, we sample $100 \times p$ datapoints from each of the 3 components. These sampled datapoints are cubed so that none of the components is normally distributed. 50 datasets are simulated for each setting. The results for different values of $p$ are given in Tables 3, 4, 5, 6 and 7. At higher values of $p$, SIA outperforms all other methods and at lower values of $p$, SIA outperforms EM and GD. When

the dimensionality is small and the cluster separation is poor, we find that the performance of Mclust is slightly better than that of SIA.

Table 3: ARI (mean and SD) on varying $\lambda$ and $p = 2$

| $\lambda$ | 3 | 4 | 5 | 7 |
|---|---|---|---|---|
| SIA | 0.021 | 0.073 | 0.135 | 0.281 |
| | (0.052) | (0.147) | (0.270) | (0.325) |
| AD-GD | 0.021 | 0.064 | 0.129 | 0.278 |
| | (0.052) | (0.146) | (0.218) | (0.334) |
| EM | 0.024 | 0.043 | 0.111 | 0.284 |
| | (0.042) | (0.060) | (0.206) | (0.355) |
| Mclust | **0.104** | **0.189** | **0.226** | **0.447** |
| | **(0.019)** | **(0.038)** | **(0.036)** | **(0.224)** |

Table 4: ARI (mean and SD) on varying $\lambda$ and $p = 3$

| $\lambda$ | 3 | 4 | 5 | 7 |
|---|---|---|---|---|
| SIA | 0.119 | 0.213 | **0.399** | **0.881** |
| | (0.081) | (0.090) | **(0.099)** | **(0.069)** |
| AD-GD | 0.105 | 0.198 | 0.381 | 0.873 |
| | (0.059) | (0.073) | (0.094) | (0.071) |
| EM | 0.075 | 0.221 | 0.395 | 0.632 |
| | (0.038) | (0.117) | (0.178) | (0.358) |
| Mclust | **0.139** | **0.228** | 0.290 | 0.731 |
| | **(0.018)** | **(0.041)** | (0.081) | (0.228) |

Table 5: ARI (mean and SD) on varying $\lambda$ and $p = 5$

| $\lambda$ | 3 | 4 | 5 | 7 |
|---|---|---|---|---|
| SIA | 0.163 | **0.504** | **0.757** | **0.926** |
| | (0.110) | **(0.129)** | **(0.103)** | **(0.036)** |
| AD-GD | 0.178 | 0.451 | 0.678 | 0.908 |
| | (0.075) | (0.088) | (0.089) | (0.048) |
| EM | 0.143 | 0.316 | 0.527 | 0.856 |
| | (0.116) | (0.195) | (0.297) | (0.254) |
| Mclust | **0.184** | 0.321 | 0.610 | 0.839 |
| | **(0.013)** | (0.154) | (0.262) | (0.169) |

Table 6: ARI (mean and SD) on varying $\lambda$ and $p = 10$

| $\lambda$ | 3 | 4 | 5 | 7 |
|---|---|---|---|---|
| SIA | **0.559** | **0.836** | **0.893** | **0.983** |
| | **(0.109)** | **(0.130)** | **(0.239)** | **(0.015)** |
| AD-GD | 0.553 | 0.773 | 0.846 | 0.949 |
| | (0.109) | (0.220) | (0.235) | (0.038) |
| EM | 0.340 | 0.477 | 0.571 | 0.835 |
| | (0.189) | (0.325) | (0.412) | (0.294) |
| Mclust | 0.265 | 0.626 | 0.727 | 0.829 |
| | (0.052) | (0.270) | (0.259) | (0.237) |

Table 7: ARI (mean and SD) on varying $\lambda$ and $p = 20$

| $\lambda$ | 3 | 4 | 5 | 7 |
|---|---|---|---|---|
| SIA | **0.850** | **0.893** | **0.947** | **0.966** |
| | **(0.162)** | **(0.172)** | **(0.469)** | **(0.441)** |
| AD-GD | 0.846 | 0.843 | 0.861 | 0.926 |
| | (0.058) | (0.279) | (0.491) | (0.441) |
| EM | 0.684 | 0.750 | 0.879 | 0.995 |
| | (0.234) | (0.345) | (0.263) | (0.005) |
| Mclust | 0.328 | 0.747 | 0.779 | 0.796 |
| | (0.008) | (0.256) | (0.250) | (0.249) |

## 5.2 Varying Covariance structures and Unbalanced mixtures

We sample from a 2 component $p$-dimensional GMM whose means are $(-0.5, \ldots, -0.5)_p$ and $(0.5, \ldots, 0.5)_p$ and covariances matrices are $\boldsymbol{\Sigma}_k = (\boldsymbol{\Sigma}_k^{1/2}(\boldsymbol{\Sigma}_k^{1/2})^T)$. The parameters of $\boldsymbol{\Sigma}_k^{1/2}$ are sampled randomly to capture different covariance structures. We demonstrate in Appendix N that this sampling approach indeed yields diverse covariance structures. The simulated datapoints are then cubed (for misspecification). Keeping the number of data points ($N_1 \times p$) in cluster 1 constant at $100 \times p$, we vary the number of datapoints ($N_2 \times p$) in cluster 2 as $\{20 \times p, 50 \times p, 100 \times p\}$. We simulate 50 different datasets for each setting. We vary the dimensionality $p$ as $\{2, 3, 5, 10, 20\}$. The results are given in Tables 8, 9 and 10. We observe that when the dimensionality is low, clustering performance is better at higher imbalance. Appendix K shows an illustration. At higher values of $p$, clustering performance is better at lower imbalance. Overall, we find that SIA performs on par or better than EM, AD-GD and MClust.

## 6 Evaluation on Real Datasets

We evaluate SIA on real datasets: (I) Wine (Forina et al., 1986), (II) IRIS (Fisher, 1936), (III) Abalone (Nash et al., 1994)), (IV) Urban Land cover (Johnson & Xie, 2013), (V) LUSC RNA-Seq (Kandoth et al., 2013)), (VI) Breast Cancer Wolberg et al. (1995), (VII) Crabs Campbell & Mahon (1974), (VIII) Wholesale Cardoso (2014), (IX) Ceramics mis (2019), and (X) HTRU Lyon (2017).

Table 8: ARI (mean and SD) for different $p$ and $N_2 = 20$

| p | 2 | 3 | 5 | 10 | 20 |
|---|---|---|---|---|---|
| SIA | **0.192** | **0.128** | **0.077** | **0.048** | 0.0311 |
| | **(0.333)** | **(0.241)** | **(0.086)** | **(0.056)** | (0.035) |
| AD-GD | 0.168 | 0.122 | 0.020 | 0.001 | 0.001 |
| | (0.328) | (0.244) | (0.079) | (0.007) | (0.01) |
| EM | 0.162 | 0.084 | 0.064 | 0.0344 | 0.030 |
| | (0.343) | (0.234) | (0.088) | (0.054) | (0.035) |
| MClust | 0.168 | 0.086 | 0.052 | 0.045 | **0.035** |
| | (0.329) | (0.181) | (0.079) | (0.028) | **(0.015)** |

Table 9: ARI (mean and SD) for different $p$ and $N_2 = 50$

| p | 2 | 3 | 5 | 10 | 20 |
|---|---|---|---|---|---|
| SIA | **0.163** | **0.117** | **0.099** | **0.086** | **0.245** |
| | **(0.259)** | **(0.170)** | **(0.102)** | **(0.055)** | **(0.215)** |
| AD-GD | 0.122 | 0.112 | 0.068 | 0.052 | 0.208 |
| | (0.259) | (0.171) | (0.101) | (0.066) | (0.242) |
| EM | 0.158 | 0.105 | 0.078 | 0.070 | 0.061 |
| | (0.258) | (0.173) | (0.095) | (0.054) | (0.061) |
| MClust | 0.154 | 0.102 | 0.052 | 0.025 | 0.017 |
| | (0.322) | (0.148) | (0.088) | (0.028) | (0.015) |

Table 10: ARI (mean and SD) for different $p$ and $N_2 = 100$

| p | 2 | 3 | 5 | 10 | 20 |
|---|---|---|---|---|---|
| SIA | 0.088 | **0.134** | **0.089** | **0.108** | 0.474 |
| | (0.073) | **(0.105)** | **(0.112)** | **(0.077)** | (0.221) |
| AD-GD | 0.066 | 0.125 | 0.081 | 0.094 | **0.538** |
| | (0.075) | (0.106) | (0.114) | (0.084) | **(0.260)** |
| EM | 0.084 | 0.108 | 0.073 | 0.089 | 0.078 |
| | (0.070) | (0.098) | (0.088) | (0.078) | (0.075) |
| MClust | **0.131** | 0.110 | 0.057 | 0.012 | 0.003 |
| | **(0.240)** | (0.087) | (0.079) | (0.014) | (0.006) |

The datasets are diverse with respect to values of $n, p$ and $K$ and such that their underlying clusters do not appear to be Gaussian or well-separated (as seen through their scatter-plots). EM, AD-GD, MClust and SIA are initialized using K-Means, with 10 different random initializations within K-Means, and the best result obtained is reported.

Table 11: Clustering performance (ARI and RI) on ten real datasets

| Dataset | n | p | K | ARI | | | | RI | | | |
|---|---|---|---|---|---|---|---|---|---|---|---|
| | | | | SIA | EM | AD-GD | MClust | SIA | EM | AD-GD | MClust |
| I | 178 | 13 | 3 | **0.63** | 0.462 | 0.375 | 0.62 | **0.835** | 0.742 | 0.721 | 0.829 |
| II | 150 | 4 | 3 | **0.92** | 0.90 | 0.90 | 0.90 | **0.965** | 0.957 | 0.957 | 0.957 |
| III | 4177 | 8 | 3 | **0.130** | 0.121 | 0.089 | 0.072 | **0.587** | 0.582 | 0.577 | 0.566 |
| IV | 168 | 148 | 9 | **0.11** | - | 0 | - | **0.613** | - | 0 | - |
| V | 130 | 206 | 2 | **0.023** | - | 0 | -0.01 | **0.513** | - | 0 | 0.496 |
| VI | 569 | 30 | 2 | **0.610** | 0.593 | 0.213 | 0.593 | **0.805** | 0.797 | 0.621 | 0.797 |
| VII | 200 | 5 | 4 | **0.822** | 0.818 | 0 | 0.322 | **0.933** | 0.931 | 0 | 0.744 |
| VIII | 400 | 6 | 6 | **0.188** | 0.116 | 0.112 | 0.132 | **0.651** | 0.626 | 0.599 | 0.548 |
| IX | 88 | 17 | 2 | 0.866 | 0.783 | 0.9101 | **1.0** | 0.933 | 0.891 | 0.955 | **1.0** |
| X | 17898 | 8 | 2 | **0.402** | 0.349 | 0.348 | 0.344 | **0.770** | 0.739 | 0.738 | 0.736 |

Table 11 shows the dataset statistics and the ARI obtained (using the provided ground truth) by each method. In datasets IV and V, the number of observations is lesser than the dimensions ($n < p$), EM fails to run and AD-GD assigns all the points to a single cluster.

## Case Study

The Wine dataset contains the results of a chemical analysis of wines, yielding 13 features, from 3 different cultivars. We use this dataset, without the labels of wine types, and fit EM, AD-GD, MClust and SIA for $K \in \{2, 3, 4\}$. The log-likelihood (LL), MPKL, KLDivs, AIC and ARI values are shown in table 12. For each $K$ we show the results for multiple runs with different initializations.

For $K = 2$, among the 3 EM solutions, while clustering S1 has better LL, its MPKL is higher (worse) than those obtained by the other clustering outputs (S2 and S3). So, even among EM solutions, by trading off likelihood for an improvement in MPKL, we can choose better clustering solutions with higher accuracy. The LL of SIA is better than that of AD-GD (which was used in step 1 of SIA) indicating that its penalty term can aid in escaping local maxima. The ARI from SIA (S5) is comparable to the best results from EM. We

Table 12: Clustering Results for Wine dataset

| S | $K$ | Algorithm | LL | AIC | KLF | KLB | MPKL | ARI |
|---|---|---|---|---|---|---|---|---|
| 1 | 2 | EM | -2983.6 | 6385.3 | 86.3 | 39.5 | 46.8 | 0.46 |
| 2 | 2 | EM | -3117.4 | 6652.9 | 17.2 | 14.8 | 2.3 | 0.57 |
| 3 | 2 | EM | -3083.0 | 6584.0 | 22.2 | 17.36 | 4.8 | 0.59 |
| 4 | 2 | AD-GD | -3081.6 | 6581.2 | 56.1 | 29.6 | 26.5 | 0.55 |
| 5 | 2 | SIA | -3072.0 | 6562.0 | 42.1 | 27.5 | 14.6 | 0.57 |
| 6 | 3 | EM | -2901.0 | 6430.0 | 321.29 | 91.44 | 229.8 | 0.46 |
| 7 | 3 | EM | -2915.7 | 6459.4 | 93.25 | 118 | 51.68 | 0.62 |
| 8 | 3 | AD-GD | -3030.04 | 6688.8 | 165.9 | 152.1 | 50.7 | 0.38 |
| 9 | 3 | SIA | -2921.45 | 6470.9 | 57.7 | 60.8 | **14.66** | **0.63** |
| 10 | 4 | EM | -2637.7 | **6113.4** | 2.86e8 | 407.7 | 134e8 | 0.52 |
| 11 | 4 | EM | -2660.7 | 6158.0 | 513e7 | 362.1 | 513e7 | 0.52 |
| 12 | 4 | AD-GD | -2910.7 | 6659.4 | 1192.4 | 364.0 | 486.65 | 0.35 |
| 13 | 4 | SIA | -2850.5 | 6539.0 | 519.0 | 304.9 | 214.05 | **0.67** |

Table 13: Results for the best model for each $K$ (S3, S9, S13 in Table 12). ①, ②, ③: 3 wine types.

| $K$ | ① | ② | ③ |
|---|---|---|---|
| 2 | 59 | | 40 |
| | | 71 | 8 |
| 3 | 56 | 6 | |
| | | 56 | 7 |
| | 3 | 9 | 41 |
| 4 | 35 | 3 | |
| | | 63 | 0 |
| | 75 | 5 | 48 |
| | 17 | | |

observe similar trends for $K = 3$. Among EM solutions (S6 and S7), the solution with better MPKL (S7) achieves a better ARI. Again, the LL of SIA (S9) is better than that of AD-GD (S8), which was used in step 1 of SIA. The best ARI is achieved by SIA which also has the lowest MPKL value. For $K = 4$, we observe that AD-GD solution (S12) does not have high LL, nor does it have a low MPKL value. When SIA step-2 is initialized with the output of EM (S13), we observe a decrease in MPKL and an improvement in ARI over both EM and AD-GD. The confusion matrices of models with the best MPKL values for each $K$ is in Table 13. Misspecification of the components is evident in the data scatterplot and S9 visually has the most well separated clusters (both in Appendix M).

# 7 Conclusion

In this paper, we illustrate and discuss the problem of inferior solutions that occur while clustering with misspecified GMMs. They differ in their characteristics from spurious solutions in terms of the asymmetry of component orientation and sizes, and frequency of occurrence. Our theoretical analysis highlights a new relation between such asymmetry of fitted components and misspecification. Further investigation of this connection would be interesting to explore in the future.

We propose a new penalty term based on the Kullback Leibler divergence between pairs of fitted components that, by design, avoids inferior solutions. We prove that the penalized likelihood is bounded and avoids degeneracy. Gradient computations for this penalized likelihood is difficult but Automatic Differentiation can be done effortlessly. We develop algorithms SIA for clustering and MPKL for model selection, and evaluate their performance, on clustering synthetic and real datasets with misspecified GMMs. Our simulation study reveals that SIA works well in several cases of misspecification that we examine, particularly when the cluster separation is low. At high cluster separation, the effect of misspecification on clustering is less acute and other methods, like MClust, also perform well, and the performance of SIA is comparable. Although the penalty term in SIA is designed to penalize component asymmetry, SIA performs well even when there are significant differences with respect to orientations and sizes of the true components in our simulation study. This is because SIA does not use hard constraints on the orientations and sizes and the functional penalization approach enables a data-dependent discovery of the underlying clusters. Nevertheless, it is possible in some cases for SIA to fit symmetric components when the underlying clusters have covariances with different orientations and sizes. The use of our proposed MPKL criterion in conjunction with other likelihood-based criteria like AIC or BIC, as discussed in our case study, can guide the user to find good clustering solutions in such cases. Statistical guarantees with this penalized likelihood approach and extensions to high-dimensional settings require further work and would be of theoretical and practical interest.

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

# Appendix

## A  Symbols and Notation

| Symbol | Meaning |
|---|---|
| $f$ | overall probability density from which data is sampled |
| $g$ | misspecified model for data sampled from $f$ |
| $f_i$ | probability density of the $i$-th component |
| $\pi_i$ | mixing proportions or weight of the $i$-th component |
| $\boldsymbol{\theta}_i$ | parameters of $i$-th mixture component |
| $\boldsymbol{\theta}$ | all parameters of the model |
| $\boldsymbol{\theta}^*$ | true parameters of the model from which data is sampled |
| $\boldsymbol{\Theta}$ | set of all possible parameters i.e. the entire parameter space |
| $\mathbf{F}_i$ | factors associated with $i$-th datapoint |
| $K$ | total of number of components in the model |
| $k, k^{'}, k_1, k_2$ | indices over the total number of components $K$ |
| $\mathbf{X}_i$ | $i$-th random datapoint |
| $\mathbf{x}_i$ | $i$-th observed datapoint |
| $\mathbf{X}$ | all the $n$ random datapoints |
| $\mathbf{x}$ | all the $n$ observed datapoints |
| $n$ | Total number of $iid$ datapoints considered |
| $p$ | Total number of dimensions or features of the data |
| $p_e$ | Total number of free parameters in the model |
| $\mathcal{L}(\boldsymbol{\theta})$ | Log-Likelihood of observing $\mathbf{x}$ if the parameters were $\boldsymbol{\theta}$ |
| $\mathcal{M}(\boldsymbol{\theta})$ | Penalized SIA Log-Likelihood if the parameters were $\boldsymbol{\theta}$ |
| $\mathcal{N}$ | density of multi-variate Gaussian distribution |
| $\mathcal{B}$ | Bernoulli distribution |
| $\boldsymbol{\mu}_i, \mu_{ij}$ | mean of $i$-th component of a GMM and its value of along $j$-th dimension respectively |
| $\boldsymbol{\Sigma}_i, \Sigma_{ijk}$ | Covariance matrix of $i$-th component of a GMM and its $jk$-th element respectively |
| $\mathbf{U}_i$ | square-root of $i$-th covariance matrix $\boldsymbol{\Sigma}_i$ |
| $\hat{a}^t$ | Estimate of parameter $a$ at the end of iteration $t$ |
| $c, C$ | Constants independent of model parameters |
| $\lambda$ | factor controlling the cluster separation in simulations |
| $\lambda_k$ | hyperparameter controlling the fitted component covariance |
| $\mathbb{I}$ | Identity matrix |
| $\epsilon$ | learning rate in the vanilla gradient descent step |
| $\gamma$ | convergence threshold for stopping criterion |
| $b$ | factor controlling the variance of the contaminating component |
| $\tau(\bar{\boldsymbol{\theta}})$ | Asymmetry coefficient for inferred GMM parameters $\bar{\boldsymbol{\theta}}$ |
| $C_2$ | constant dependent on the true model parameters |
| $D, D'$ | Discrete measures |
| $\Pi(D, D')$ | Set of all Couplings between $D, D'$ |
| $h$ | a coupling between $D, D'$ |
| $G^*$ | 3-Component corrupted GMM |
| $G$ | 2-Component GMM being fit to $G^*$ |

Table 14: Symbols used in the paper

## B  Spurious Solutions

Spurious solutions are local maximizers of the likelihood function but lack real-life interpretability and hence do not provide a good clustering of the data. It is a consequence of the unboundedness of the GMM likelihood function for unrestricted component covariance matrices. As discussed in McLachlan & Peel (2000), spurious solutions may be obtained when:

- a fitted component has very small non-zero variance for univariate data or generalized covariance, i.e., the determinant of the covariance matrix, for multivariate data. Such a component corresponds to a cluster containing very few data points close together or, for multivariate data, in a lower dimensional subspace.

- the model fits a small localized random pattern in the data instead of an underlying group structure. Such solutions have very few points in one cluster with little variation, compared to other clusters, in the cluster's axes, or for multivariate data, small eigenvalues for the component covariance matrix.

- where the likelihood increases by fitting the covariance matrix of a component on just one or a few datapoints distant from the other samples.

In some cases, e.g., when a component is fitted over very few datapoints, spurious solutions lead to singularities in the component covariance matrices that can be detected during EM inference. There are other cases as well, when the parameters lie close to the boundary of the parameter space, when the component covariance matrices are not singular but may be close to singular for some components.

McLachlan & Peel (2000) observed that convergence to spurious solutions happen rarely; they are dependent on initialization – often occurring only for some initializations; and convergence to non-spurious global maxima becomes difficult with increasing dimensionality.

## C  Model Selection

Information Criterion-based model selection methods are designed to favor parsimony in modeling, by penalizing overfitting of the model(Konishi & Kitagawa, 2008). Common likelihood-based criteria include Akaike Information Criterion (AIC):

$$\text{AIC} := 2p_e - 2\mathcal{L}(\hat{\boldsymbol{\theta}}),$$

and Bayesian Information Criterion (BIC):

$$\text{BIC} := p_e \ln(n) - 2\mathcal{L}(\hat{\boldsymbol{\theta}}),$$

where $p_e$ is the number of parameters to be estimated in the model.

Traditional likelihood based model selection criteria such AIC and BIC cannot used reliably when there is misspecification as they assume that the model has been rightly specified (Konishi & Kitagawa, 2008; Lv & Liu, 2014). There exists more generalized information criteria such as Takeuchi Information Criteria Takeuchi (1976) which can accomodate misspecified models; however their practical use is limited by the requirement to evaluate Hessians with respect to model parameters. It has been pointed out that MLE based criteria such as AIC and BIC are based on asymptotic normality and hence, cannot be used for model selection in high-dimensions where $n << p$ (Amari et al., 2006; Akaho & Kappen, 2000; Giraud, 2014). Moreover, in high dimensions, it has been observed that BIC tends to underestimate the number of components whereas AIC overestimates the number of components (Melnykov et al., 2010).

# D Automatic Differentiation based Gradient Descent

The contents of this section are from our previous work Kasa & Rajan (2022), with examples from Baydin et al. (2018).

## D.1 Gradient Descent and Automatic Differentiation

Automatic Differentiation (AD) (also known as algorithmic differentiation) is a suite of computational techniques for efficient and accurate evaluation of derivatives of numeric functions. In traditional GD based inference, gradients are required in closed form which becomes laborious or intractable to derive as the model complexity increases, e.g., for MFA. To automate the computation of derivatives three classes of techniques have been developed: (a) Finite Differentiation (FD) (b) Symbolic Differentiation (SD) and (c) Automatic or Algorithmic Differentiation (AD). Although easy to code, FD is slow at high dimensions and susceptible to floating point errors. SD provides exact symbolic expressions of derivatives but has high computational complexity, in both time and memory and cannot be used when functions are defined using programmatic constructs such as conditions and loops. The complexity and errors in both FD and SD increase for the computation of higher derivatives and partial derivatives of vector-valued functions. AD overcomes all these limitations of FD and SD. It provides efficient and accurate numerical evaluation of derivatives without requiring closed form expressions.

The numerical computation of a function can be decomposed into a finite set of elementary operations. These operations most commonly include arithmetic operations and transcendental function evaluations. The key idea of AD is to numerically compute the derivative of a function by combining the derivatives of the elementary operations through the systematic application of the chain rule of differential calculus. The efficiency of the computation is improved by storing evaluated values of intermediate sub-expressions that may be re-used. Backpropagation, used to train neural networks, is a specific form of AD which is more widely applicable. We refer the reader to recent surveys (Baydin et al., 2018; Margossian, 2019) for more details.

Efficient implementations of AD are available in several programming languages and frameworks, e.g. Python (Maclaurin et al., 2015), R (Pav, 2016), Pytorch (Paszke et al., 2017) and Stan (Carpenter et al., 2015). Many first and second order gradient-based optimization algorithms are implemented in these libraries. In this paper, we use Adam, a first order method that computes individual adaptive learning rates for different parameters from estimates of first and second moments of the gradients (Kingma & Ba, 2015).

## D.2 Illustration of Automatic Differentiation

In symbolic differentiation, we first evaluate the complete expression and then differentiate it using rules of differential calculus. First note that a naive computation may repeatedly evaluate the same expression multiple times, e.g., consider the rules:

$$\frac{d}{dx}\left(F(x) + G(x)\right) \rightsquigarrow \frac{d}{dx}F(x) + \frac{d}{dx}G(x) \tag{3}$$

$$\frac{d}{dx}\left(F(x)\,g(x)\right) \rightsquigarrow \left(\frac{d}{dx}F(x)\right)G(x) + F(x)\left(\frac{d}{dx}G(x)\right) \tag{4}$$

Let $H(x) = F(x)G(x)$. Note that $H(x)$ and $\frac{d}{dx}H(x)$ have in common: $F(x)$ and $G(x)$, and on the right hand side, $F(x)$ and $\frac{d}{dx}F(x)$ appear separately. In symbolic differentiation we plug the derivative of $F(x)$ and thus have nested duplications of any computation that appears in common between $F(x)$ and $\frac{d}{dx}F(x)$. In this manner symbolic differentiation can produce exponentially large symbolic expressions which take correspondingly long to evaluate. This problem is called **expression swell**. To illustrate the problem, consider the following iterations of the logistic map $l_{n+1} = 4l_n(1 - l_n)$, $l_1 = x$ and the corresponding derivatives of $l_n$ with respect to $x$. Table 15 clearly shows that the number of repetitive evaluations increase with $n$.

If the symbolic form is not required and only numerical evaluation of derivatives is required, computations can be simplified by storing the values of intermediate sub-expressions. Further efficiency gains in computation can

Table 15: Iterations of the logistic map $l_{n+1} = 4l_n(1 - l_n)$, $l_1 = x$ and the corresponding derivatives of $l_n$ with respect to $x$, illustrating expression swell (from Baydin et al. (2018))

| $n$ | $l_n$ | $\frac{d}{dx}l_n$ | $\frac{d}{dx}l_n$ (Simplified form) |
|---|---|---|---|
| 1 | $x$ | 1 | 1 |
| 2 | $4x(1-x)$ | $4(1-x) - 4x$ | $4 - 8x$ |
| 3 | $16x(1-x)(1-2x)^2$ | $16(1-x)(1-2x)^2 - 16x(1-2x)^2 - 64x(1-x)(1-2x)$ | $16(1 - 10x + 24x^2 - 16x^3)$ |
| 4 | $64x(1-x)(1-2x)^2 (1-8x+8x^2)^2$ | $128x(1 - x)(-8 + 16x)(1-2x)^2(1-8x+8x^2) + 64(1-x)(1-2x)^2(1-8x+8x^2)^2 - 64x(1-2x)^2(1-8x+8x^2)^2 - 256x(1 - x)(1-2x)(1-8x+8x^2)^2$ | $64(1 - 42x + 504x^2 - 2640x^3 + 7040x^4 - 9984x^5 + 7168x^6 - 2048x^7)$ |

be achieved by interleaving differentiation and simplification steps. The derivative of $l_{n+1} = 4l_n(1-l_n)$ can be found using the chain rule $\frac{dl_{n+1}}{dl_n} \frac{dl_n}{dl_{n-1}} \ldots \frac{dl_1}{dl_x}$ which simplifies to $4(1-l_n-l_n)4(1-l_{n-1}-l_{n-1})\ldots 4(1-x-x)$. Note that evaluation in AD is computationally linear in $n$ (because we add only one $(1 - l_n - l_n)$ for each increase by 1). This linear time complexity is achieved due to 'carry-over' of the derivatives at each step, rather than evaluating the derivative at the end and substituting the value of x.

The Python code below shows the simplicity of the implementation for this problem.

```
from autograd import grad
def my_func(x,n):
    p = x
    y = x * (1 - x)
    for i in range(n):
        y = y*(1 - y)

    return y
grad_func = grad(my_func)
grad_func(0.5,4)
```

Consider the following recursive expressions: $l_0 = \frac{1}{1+e^x}$, $l_1 = \frac{1}{1+e^{l_0}}$, ....., $l_n = \frac{1}{1+e^{l_{n-1}}}$ We evaluate the derivative of $l_n$ with respect to $x$ and compare the runtime in Mathematica (SD) vs PyTorch (AD) for various values of $n$. As $n$ increases, it is expected that runtime also increases. It can be seen from the results in Table 16 that runtime increases linearly for AD (using PyTorch) whereas it increases exponentially for SD (using Mathematica).

Table 16: Average runtime (over 1000 runs)

| n | AD | SD |
|---|------|------|
| 1 | 0.00013 | 0.00000 |
| 5 | 0.00030 | 0.00005 |
| 10 | 0.00051 | 0.00023 |
| 50 | 0.00293 | 0.00437 |
| 100 | 0.00433 | 0.15625 |
| 200 | 0.00917 | 1.45364 |

### D.3   Reparametrizations for AD-GD inference on GMMs

As mentioned in section 2, EM solves three main problems in GMM inference. In AD-GD, *Problem 1* is solved inherently because we do not need to express the gradients in closed form by virtue of using AD. Further, the second-order Hessian matrix may also be evaluated using AD, enabling us to use methods with faster convergence. To tackle *Problem 2*, instead of gradients with respect to $\mathbf{\Sigma}_k$, we compute the gradients with respect to $\mathbf{U}_k$, where $\mathbf{\Sigma}_k = \mathbf{U}_k\mathbf{U}_k^T$. We first initialize $\mathbf{U}_k$ as identity matrices. Thereafter, we keep adding the gradients to the previous estimates of $\hat{\mathbf{U}}_k^{t+1}$, i.e.

$$\hat{\mathbf{U}}_k^{t+1} := \hat{\mathbf{U}}_k^t + \epsilon\frac{\partial\mathcal{L}}{\partial\mathbf{U}_k}; \ \ \hat{\mathbf{\Sigma}}_k^{t+1} := \hat{\mathbf{U}}_k^{t+1}\hat{\mathbf{U}}_k^{t+1^T} \tag{5}$$

where $\epsilon$ is the learning rate and superscripts $t, t+1$ denote iterations in GD. If the gradients are evaluated with respect to $\mathbf{\Sigma}_k$ directly, there is no guarantee that updated $\hat{\mathbf{\Sigma}}_k^{t+1} = \hat{\mathbf{\Sigma}}_k^t + \epsilon\frac{\partial\mathcal{L}}{\partial\mathbf{\Sigma}_k}$ will still remain PD. However, if the gradients are evaluated with respect to $\mathbf{U}_k$, by construction $\hat{\mathbf{\Sigma}}_k^{t+1}$ will always remains PD. Cholesky decomposition for reparameterizing $\mathbf{\Sigma}_k$ can also be used (Salakhutdinov et al., 2003). *Problem 3* is solved by using the log-sum-exp trick (Robert, 2014). We start with unbounded $\alpha_k$ as the log-proportions: $\log \pi_k = \alpha_k - \log(\sum_{k'=1}^K e^{\alpha_{k'}})$. We need not impose any constraints on $\alpha_k$ as the final computation of $\pi_k$ automatically leads to normalization, because $\pi_k = \frac{e^{\alpha_k}}{\sum_{k'=1}^K e^{\alpha_{k'}}}$. Therefore, we reparametrize a constrained optimization problem into an unconstrained one (without using Lagrange multipliers) and we can update $\hat{\pi}_k$ as follows:

$$\hat{\alpha}_k^{t+1} := \hat{\alpha}_k^t + \epsilon\frac{\partial\mathcal{L}}{\partial\alpha_k}; \ \ \hat{\pi}_k^{t+1} := \frac{e^{\hat{\alpha}_k^{t+1}}}{\sum_{k'=1}^K e^{\hat{\alpha}_{k'}^{t+1}}} \tag{6}$$

# E    Performance of EM and AD-GD on Misspecified GMM: Empirical Results

## E.1    EM and AD-GD: A comparison of clustering performance

We compare the clustering performance of AD-GD and EM on pure GMM data as well as non-GMM (misspecified) data. We do so by simulating 3600 datasets as described below and running both the algorithms with matching settings with respect to random initialization, number of iterations (maximum of 100 iterations) and convergence threshold (1e-5). The details of the simulations and results are given below.

**Pure GMM data**   : We simulate datasets from 36 different 2–dimensional 3–component Gaussian Mixture Models that differ in their means and covariance matrices. This is to test different types of data - from well-separated components to highly overlapping components. The GMM parameters are varied as follows:

- First, we construct a $2 \times 1$ vector $\mathbf{v}_k$ and a $2 \times 2$ matrix $\mathbf{Z}_k$ for each of these 3 components, where each element of the vectors and matrices are sampled from a standard normal distribution.

- The mean vector of the $k^{th}$ component is obtained by multiplying $\mathbf{v}_k$ by a factor $k_{\boldsymbol{\mu}}$ i.e. $\boldsymbol{\mu}_k = k_{\boldsymbol{\mu}}\mathbf{v}_k$. This factor $k_{\boldsymbol{\mu}}$ is varied from 0.25 to 1.5 in steps of 0.25, i.e., the factor $k_{\boldsymbol{\mu}}$ takes one of the six values in $\{0.25, 0.5, 0.75, 1, 1.25, 1.5\}$. Higher the value of $k_{\boldsymbol{\mu}}$, more the separation between the means of components.

- The $\mathbf{Z}_k$ vector is added to the $2 \times 2$ Identity matrix $\mathbb{I}$ (This step ensures the PD of the covariance matrix that is computed in the later steps).

- We multiply the matrix $\mathbf{Z}_k + \mathbb{I}$ by $k_{\boldsymbol{\Sigma}}$ to obtain $\mathbf{U}_k$. The scaling factor $k_{\boldsymbol{\Sigma}}$ is varied from 0.05 to 0.65 in steps of 0.1, i.e., the scaling factor takes one of the six values in $\{0.05, 0.15, 0.25, 0.35, 0.45, 0.55\}$. The covariance matrix of the $k^{\text{th}}$ component is $\boldsymbol{\Sigma}_k = \mathbf{U}_k\mathbf{U}_k^T$. Higher the scaling factor $k_{\boldsymbol{\Sigma}}$, more the inter-component overlap.

By choosing six different values of $k_{\boldsymbol{\mu}}$ and $k_{\boldsymbol{\Sigma}}$ each, we simulate 36 (6x6) GMM parameters with varying cluster separation. For each these 36 GMM parameters, we simulate 50 different datasets each containing 300 samples, for a total of 1800 datasets. Both the methods (EM and AD-GD) are run on each dataset with 50 random initializations and average ARI is computed for each combination of $(k_{\boldsymbol{\mu}}, k_{\boldsymbol{\Sigma}})$.

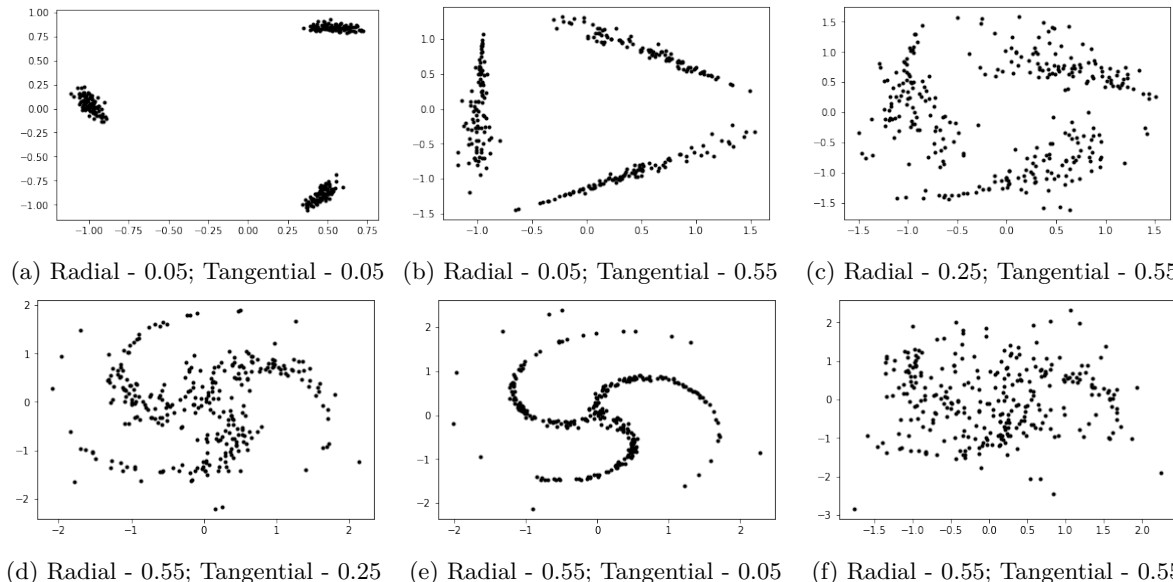

(a) Radial - 0.05; Tangential - 0.05   (b) Radial - 0.05; Tangential - 0.55   (c) Radial - 0.25; Tangential - 0.55

(d) Radial - 0.55; Tangential - 0.25   (e) Radial - 0.55; Tangential - 0.05   (f) Radial - 0.55; Tangential - 0.55

Figure 4: Data with completely different shapes can be simulated by varying the tangential and radial parameters of pinwheel dataset.

**Misspecified datasets:** The pinwheel data is generated by sampling from Gaussian distributions and then stretching and rotating the data in a controlled manner. The centers are equidistant around the unit circle. The variance is controlled by two parameters $r$ and $t$, the radial standard deviation and the tangential standard deviation respectively. The warping is controlled by a third parameter, $s$, the rate parameter. A datapoint $(x, y)$ belonging to component or arm $k$ is generated as follows:

$$(x', y') \sim \mathcal{N}(0, \mathbb{I}) \tag{7}$$
$$(x, y) = ((rx' + 1)\cos\theta_k + ty'\sin\theta_k, -(rx' + 1)\sin\theta_k + ty'\cos\theta_k), \tag{8}$$
$$\text{where} \quad \theta_k = k\frac{2\pi}{K} + s \times e^{rx'+1}$$

The MATLAB [2] and Python[3] codes for generating pinwheel datasets are available online. We simulated 36 different pinwheel configurations by varying the radial and tangential components for 3-component mixture models using the Autograd package in Python. The rate parameter $s$ of warping is fixed at 0.4 for all the 36 different combinations. This is to test multiple combinations – clusters with heavy tails to warped mixtures as shown in figure 4. The parameters of pinwheel datasets are chosen as follows:

- The radial parameter $r$ is chosen to be one of the six values $\{0.05, 0.15, 0.25, 0.35, 0.45, 0.55\}$

- The tangent parameter $t$ is chosen to be one of the six values $\{0.05, 0.15, 0.25, 0.35, 0.45, 0.55\}$

By choosing different values of $r$ and $t$, we simulate 36 (6x6) set of pinwheel parameters. For each these 36 pinwheel parameters, we simulate 50 different datasets each with 300 samples, for a total of 1800 datasets. Please refer to figure 4 to see the various kinds of component shapes and separations that have been generated.

The tables below show the difference in average ARI between clusterings obtained by EM and those from AD-GD on GMM (table 17) and Pinwheel (table 18) datasets. The remaining tables (Tables 22 –29) show the mean and standard deviations over the GMM and Pinwheel datasets for both EM and AD-GD. From the results in tables 17 and 18, we observe that in both the cases (misspecification and nomisspecification), EM inference outperforms AG-GD inference. Alexandrovich (2014)'s work on exact Newton's method using analytical derivatives also report similar results for non misspecified cases.

Table 17: Difference in average ARI of GMM datasets

| $k_{\boldsymbol{\mu}} \mid k_{\boldsymbol{\Sigma}}$ | 0.05 | 0.15 | 0.25 | 0.35 | 0.45 | 0.55 |
|---|---|---|---|---|---|---|
| 0.25 | 0.56 | 0.52 | 0.44 | 0.40 | 0.37 | 0.36 |
| 0.5 | 0.43 | 0.55 | 0.49 | 0.44 | 0.41 | 0.40 |
| 0.75 | 0.36 | 0.51 | 0.52 | 0.48 | 0.46 | 0.42 |
| 1 | 0.36 | 0.48 | 0.55 | 0.53 | 0.50 | 0.50 |
| 1.25 | 0.30 | 0.47 | 0.55 | 0.58 | 0.54 | 0.54 |
| 1.5 | 0.33 | 0.51 | 0.58 | 0.60 | 0.61 | 0.56 |

Table 18: Difference in average ARI of Pinwheel datasets

| $r \mid t$ | 0.05 | 0.15 | 0.25 | 0.35 | 0.45 | 0.55 |
|---|---|---|---|---|---|---|
| 0.05 | 0.18 | 0.30 | 0.41 | 0.50 | 0.51 | 0.08 |
| 0.15 | 0.47 | 0.50 | 0.56 | 0.58 | 0.57 | 0.24 |
| 0.25 | 0.51 | 0.58 | 0.61 | 0.62 | 0.61 | 0.47 |
| 0.35 | 0.30 | 0.39 | 0.45 | 0.53 | 0.49 | 0.43 |
| 0.45 | 0.12 | 0.17 | 0.20 | 0.24 | 0.27 | 0.30 |
| 0.55 | 0.05 | 0.08 | 0.12 | 0.16 | 0.18 | 0.19 |

---

[2]https://github.com/duvenaud/warped-mixtures/blob/master/data/pinwheel.m
[3]https://github.com/HIPS/autograd/blob/master/examples/data.py

Table 19: Average ARI of GMM datasets using EM

| $k_{\boldsymbol{\mu}} \mid k_{\boldsymbol{\Sigma}}$ | 0.05 | 0.15 | 0.25 | 0.35 | 0.45 | 0.55 |
|---|---|---|---|---|---|---|
| 0.25 | 0.78 | 0.63 | 0.52 | 0.46 | 0.42 | 0.41 |
| 0.5 | 0.81 | 0.79 | 0.68 | 0.59 | 0.53 | 0.51 |
| 0.75 | 0.84 | 0.84 | 0.78 | 0.70 | 0.64 | 0.57 |
| 1 | 0.84 | 0.85 | 0.83 | 0.77 | 0.70 | 0.67 |
| 1.25 | 0.84 | 0.86 | 0.84 | 0.82 | 0.75 | 0.71 |
| 1.5 | 0.83 | 0.86 | 0.86 | 0.83 | 0.81 | 0.74 |

Table 20: Average ARI of GMM datasets using AD-GD

| $k_{\boldsymbol{\mu}} \mid k_{\boldsymbol{\Sigma}}$ | 0.05 | 0.15 | 0.25 | 0.35 | 0.45 | 0.55 |
|---|---|---|---|---|---|---|
| 0.25 | 0.23 | 0.11 | 0.08 | 0.06 | 0.06 | 0.05 |
| 0.5 | 0.38 | 0.24 | 0.18 | 0.15 | 0.12 | 0.11 |
| 0.75 | 0.47 | 0.33 | 0.26 | 0.21 | 0.18 | 0.16 |
| 1 | 0.48 | 0.37 | 0.28 | 0.24 | 0.20 | 0.17 |
| 1.25 | 0.54 | 0.39 | 0.29 | 0.24 | 0.21 | 0.17 |
| 1.5 | 0.50 | 0.35 | 0.28 | 0.23 | 0.20 | 0.18 |

Table 21: Average ARI of Pinwheel datasets using EM

| $r \mid t$ | 0.05 | 0.15 | 0.25 | 0.35 | 0.45 | 0.55 |
|---|---|---|---|---|---|---|
| 0.05 | 0.95 | 1.00 | 1.00 | 1.00 | 0.95 | 0.48 |
| 0.15 | 0.95 | 0.97 | 1.00 | 0.98 | 0.93 | 0.58 |
| 0.25 | 0.88 | 0.95 | 0.95 | 0.94 | 0.91 | 0.74 |
| 0.35 | 0.60 | 0.69 | 0.73 | 0.78 | 0.72 | 0.64 |
| 0.45 | 0.36 | 0.41 | 0.41 | 0.44 | 0.45 | 0.46 |
| 0.55 | 0.23 | 0.26 | 0.28 | 0.31 | 0.32 | 0.32 |

Table 22: Average ARI of Pinwheel datasets using AD-GD

| $r \mid t$ | 0.05 | 0.15 | 0.25 | 0.35 | 0.45 | 0.55 |
|---|---|---|---|---|---|---|
| 0.05 | 0.77 | 0.70 | 0.59 | 0.50 | 0.44 | 0.40 |
| 0.15 | 0.48 | 0.47 | 0.44 | 0.40 | 0.36 | 0.34 |
| 0.25 | 0.37 | 0.37 | 0.34 | 0.32 | 0.29 | 0.27 |
| 0.35 | 0.30 | 0.30 | 0.28 | 0.25 | 0.23 | 0.21 |
| 0.45 | 0.24 | 0.24 | 0.22 | 0.20 | 0.18 | 0.16 |
| 0.55 | 0.19 | 0.18 | 0.17 | 0.15 | 0.14 | 0.12 |

Table 23: Std. Deviation of ARI of GMM datasets using EM

| $k_{\boldsymbol{\mu}} \mid k_{\boldsymbol{\Sigma}}$ | 0.05 | 0.15 | 0.25 | 0.35 | 0.45 | 0.55 |
|---|---|---|---|---|---|---|
| 0.25 | 0.30 | 0.27 | 0.25 | 0.24 | 0.22 | 0.23 |
| 0.5 | 0.29 | 0.25 | 0.26 | 0.27 | 0.26 | 0.25 |
| 0.75 | 0.26 | 0.23 | 0.25 | 0.26 | 0.26 | 0.27 |
| 1 | 0.25 | 0.23 | 0.22 | 0.26 | 0.26 | 0.26 |
| 1.25 | 0.25 | 0.24 | 0.23 | 0.22 | 0.26 | 0.26 |
| 1.5 | 0.24 | 0.23 | 0.22 | 0.23 | 0.22 | 0.26 |

Table 24: Std. Deviation of ARI of GMM datasets using AD-GD

| $k_{\boldsymbol{\mu}} \mid k_{\boldsymbol{\Sigma}}$ | 0.05 | 0.15 | 0.25 | 0.35 | 0.45 | 0.55 |
|---|---|---|---|---|---|---|
| 0.25 | 0.21 | 0.13 | 0.09 | 0.08 | 0.07 | 0.05 |
| 0.5 | 0.26 | 0.21 | 0.17 | 0.15 | 0.12 | 0.10 |
| 0.75 | 0.25 | 0.23 | 0.21 | 0.19 | 0.16 | 0.14 |
| 1 | 0.28 | 0.23 | 0.21 | 0.19 | 0.17 | 0.15 |
| 1.25 | 0.29 | 0.24 | 0.21 | 0.18 | 0.16 | 0.15 |
| 1.5 | 0.27 | 0.25 | 0.23 | 0.20 | 0.18 | 0.16 |

Table 25: Std. Deviation of ARI of Pinwheel datasets using EM

| $r \mid t$ | 0.05 | 0.15 | 0.25 | 0.35 | 0.45 | 0.55 |
|---|---|---|---|---|---|---|
| 0.05 | 0.15 | 0.00 | 0.00 | 0.00 | 0.16 | 0.32 |
| 0.15 | 0.15 | 0.12 | 0.00 | 0.08 | 0.15 | 0.34 |
| 0.25 | 0.12 | 0.10 | 0.10 | 0.08 | 0.06 | 0.25 |
| 0.35 | 0.15 | 0.16 | 0.17 | 0.13 | 0.18 | 0.17 |
| 0.45 | 0.13 | 0.12 | 0.16 | 0.19 | 0.18 | 0.16 |
| 0.55 | 0.08 | 0.08 | 0.09 | 0.11 | 0.12 | 0.11 |

Table 26: Std. Deviation of ARI of Pinwheel datasets using AD-GD

| $r \mid t$ | 0.05 | 0.15 | 0.25 | 0.35 | 0.45 | 0.55 |
|---|---|---|---|---|---|---|
| 0.05 | 0.03 | 0.04 | 0.04 | 0.04 | 0.05 | 0.05 |
| 0.15 | 0.04 | 0.04 | 0.05 | 0.04 | 0.05 | 0.04 |
| 0.25 | 0.04 | 0.04 | 0.04 | 0.04 | 0.04 | 0.04 |
| 0.35 | 0.03 | 0.03 | 0.04 | 0.04 | 0.03 | 0.03 |
| 0.45 | 0.03 | 0.03 | 0.03 | 0.03 | 0.03 | 0.03 |
| 0.55 | 0.03 | 0.03 | 0.03 | 0.03 | 0.03 | 0.03 |

### E.2   Summary statistics of solutions in sets 1–4 for EM

Table 27: Summary statistics of EM clustering solutions over 100 random initializations on the Pinwheel dataset (shown in fig. 1), grouped into 4 sets based on AIC ranges. Mean and standard deviation of AIC, ARI, component weights and covariance determinants, over solutions in each set.

| Set | AIC Range | AIC | ARI | $\pi_1$ | $\pi_2$ | $\pi_3$ | $|\mathbf{\Sigma}_1|$ | $|\mathbf{\Sigma}_2|$ | $|\mathbf{\Sigma}_3|$ |
|---|---|---|---|---|---|---|---|---|---|
| 1 | 771-773 | 771.9 (0.0005) | 0.625 (3e-16) | 0.257 (5e-4) | 0.265 (7e-5) | 0.477 (1e-4) | 0.0002 (4e-7) | 0.0005 (1e-7) | 0.125 (1e-4) |
| 2 | 781-782 | 781.1 (2e-6) | 0.912 (0) | 0.306 (4e-6) | 0.341 (3e-6) | 0.352 (7e-6) | 7e-4 (1e-7) | 0.01 (4e-7) | 0.01 (2e-6) |
| 3 | 786-787 | 786.8 (0.001) | 0.652 (0.002) | 0.257 (5e-5) | 0.266 (4e-4) | 0.475 (4e-4) | 2e-4 2e-7 | 5e-4 (6e-6) | 0.157 (5e-4) |
| 4 | 788-850 | 810.7 (10.84) | 0.840 (0.084) | 0.287 (0.022) | 0.319 (0.009) | 0.393 (0.03) | 3e-3 (1e-3) | 0.04 (0.02) | 0.84 (0.084) |

# F    Additional Motivating Examples of Inferior Clusterings

### F.1    Contamination with Student's T Distribution

As another motivating example, we illustrate the case where misspecification arises due to contamination by alternative distribution (Punzo & McNicholas, 2016). Consider a 2-dimensional GMM with 4 components, each with a unit spherical covariance. The means of the four components are set as given in Table 28 with $\beta = 3$. From each Gaussian component, we sample 50 datapoints. Next, we contaminate the sampled points

Table 28: Component (C) Means

|         | C-1      | C-2     | C-3      | C-4     |
|---------|----------|---------|----------|---------|
| $\mu_1$ | $-\beta$ | 0       | 0        | $\beta$ |
| $\mu_2$ | 0        | $\beta$ | $-\beta$ | 0       |

using a 2-dimensional 4-component multivariate $t$ distribution with same means and covariance structures. From each multivariate $t$ component, we sample an additional 50 datapoints. Thus each component has 100 samples and a dataset has 400 samples. We thus have samples from a Gaussian mixture contaminated by a Student $t$ mixture, where the degree of contamination can varied by the Degrees of Freedom (D.o.F) of the multivariate $t$ distribution. When the degree of freedom is high, the contamination is negligible. When the degree of freedom is low, the components based on $t$-distribution will have heavier tails, and thus the contamination is higher. By varying the degrees of freedom we can control the contamination and hence the misspecification.

We fit a 4-component GMM using EM (we obtain similar results using AD-GD as well). As seen from Fig. 5, the components tend to overlap with increasing misspecification (decreasing D.o.F). We observe that when the misspecification is high, some of the fitted components tend to have higher covariance compared to other components. We repeat the same experiment of corrupting the GMM with a multivariate $t$-distribution but with increasing the cluster separation by setting $\beta = 5$. The results are shown in Fig. 6 As seen, increasing cluster separation mitigates the affect of misspecification. A similar experiment on contamination with random noise is shown in Appendix F.2.

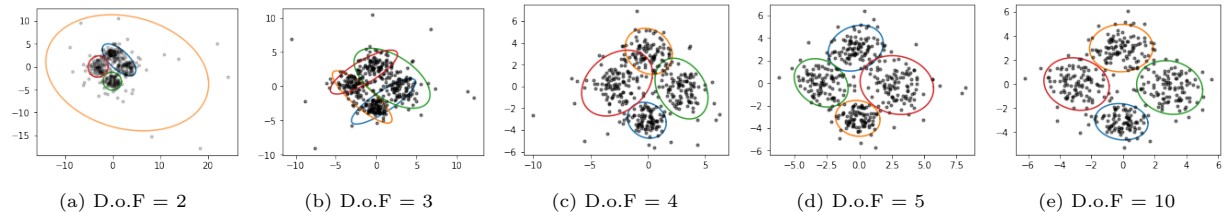

(a) D.o.F = 2    (b) D.o.F = 3    (c) D.o.F = 4    (d) D.o.F = 5    (e) D.o.F = 10

Figure 5: Illustration of GMM contaminated by Student $t$ mixture with varying degree of freedom (D.o.F) at low cluster separation. Lower the D.o.F., higher the misspecification, and greater is the difference in orientation and size of the components fitted.

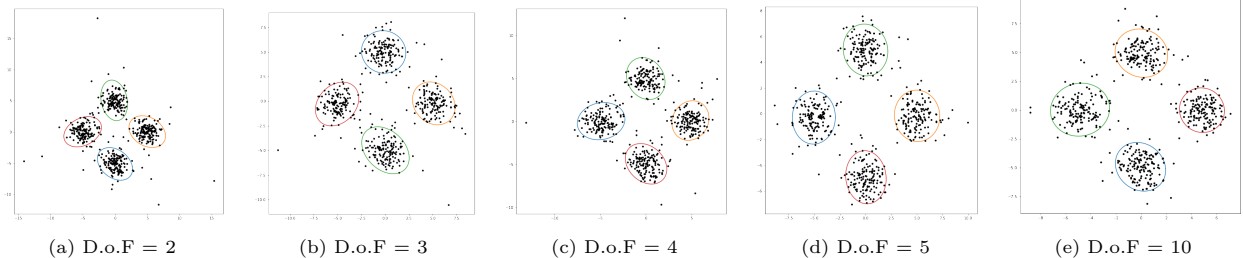

(a) D.o.F = 2    (b) D.o.F = 3    (c) D.o.F = 4    (d) D.o.F = 5    (e) D.o.F = 10

Figure 6: Illustration of GMM contaminated by Student $t$ mixture with varying degree of freedom (D.o.F) at high cluster separation. Misspecification does not seem to affect the fitted components much, only the orientation of the fitted components is affected.

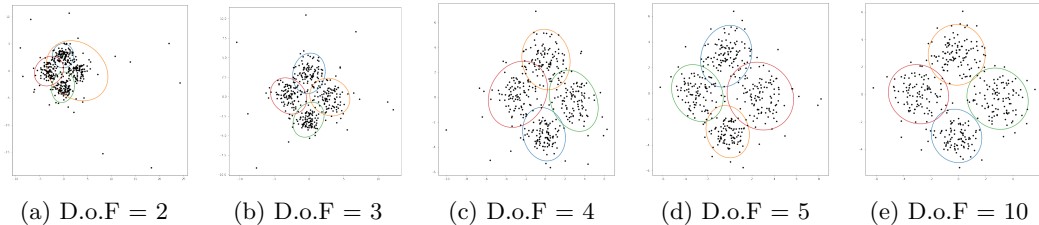

(a) D.o.F = 2     (b) D.o.F = 3     (c) D.o.F = 4     (d) D.o.F = 5     (e) D.o.F = 10

Figure 7: SIA improves the clustering performance when the Gaussian components are corrupted by a multivariate t-distribution - Refer to table 29

Table 29: Improvement in clustering performance before and after SIA step-2 for the contaminated case

| DoF | Before SIA Step 2 | | | | | After SIA Step 2 | | | | |
|---|---|---|---|---|---|---|---|---|---|---|
| | LL | KLF | KLB | MPKL | ARI | LL | KLF | KLB | MPKL | ARI |
| 2 | -1837.6 | 158.4 | 60.7 | 58.6 | 0.57 | -1943.9 | 36.8 | 41.6 | 8.8 | 0.76 |
| 3 | -1813.4 | 57.2 | 80.9 | 16.6 | 0.37 | -1871.0 | 48.2 | 46.2 | 4.3 | 0.79 |
| 4 | -1725.8 | 48.6 | 89.1 | 11.7 | 0.75 | -1735.1 | 39.6 | 54.3 | 3.8 | 0.78 |
| 5 | -1683.9 | 67.1 | 82.9 | 11.12 | 0.79 | -1694.5 | 46.1 | 54.6 | 4.9 | 0.82 |
| 10 | -1640.7 | 68.0 | 96.9 | 13.9 | 0.87 | -1648.3 | 56.3 | 65.8 | 4.2 | 0.87 |

### F.2 Contamination using uniform noise

We construct a four component GMM with unit covariances and means $\{(-3, 0), (0, 3), (0, -3), (3, 0)\}$. We sample 50 datapoints from each component. We add datapoints sampled uniformly from the square $(-6, 6) \times (-6, 6)$ as noise. We fit a 4-component GMM using EM in each case. Fig. 8 shows that as we increase the number of noisy datapoints added to the our original data sampled from GMM, the adverse effect of misspecification on EM increases. Fig. 10 shows that SIA performs well and obtains good clustering. Table 30 shows the loglikelihood, KLF, KLB, MPKL and ARI values for each case, before and after step 2 of SIA. We repeat the same experiment of adding noisy data samples, but with increased cluster separation by choosing the means as $\{(-5, 0), (0, 5), (0, -5), (5, 0)\}$. Fig. 9 shows that increasing the cluster separation mitigates the effect of misspecification with EM.

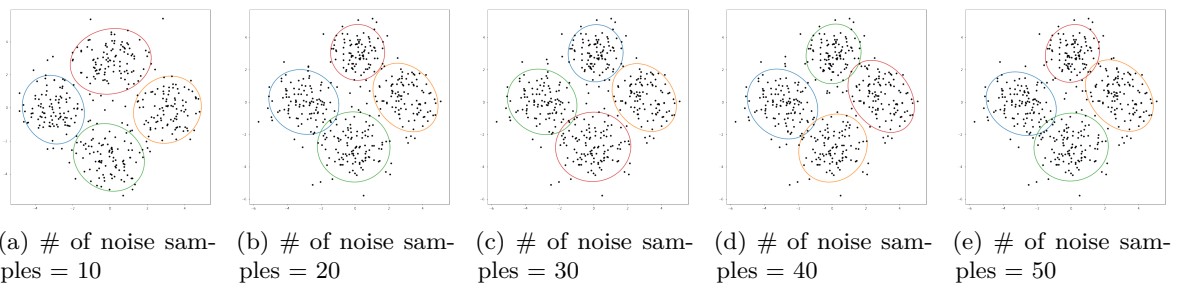

(a) # of noise samples = 10    (b) # of noise samples = 20    (c) # of noise samples = 30    (d) # of noise samples = 40    (e) # of noise samples = 50

Figure 8: As the number of noisy samples increases, the affect of misspecification also increases

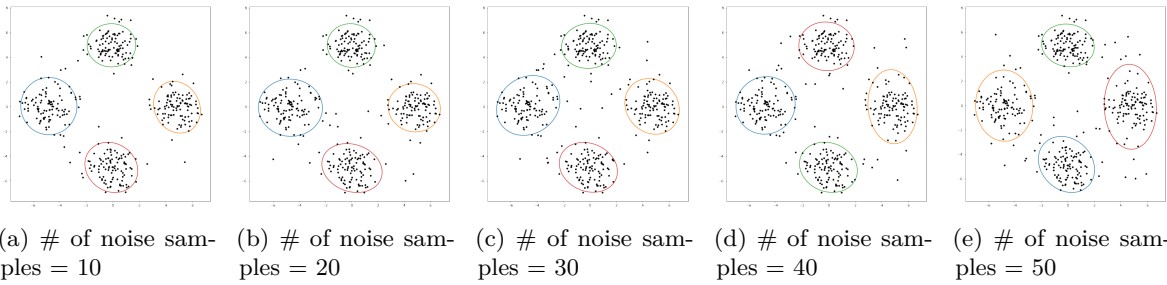

(a) # of noise samples = 10

(b) # of noise samples = 20

(c) # of noise samples = 30

(d) # of noise samples = 40

(e) # of noise samples = 50

Figure 9: Increasing the cluster separation mitigates the effect of uniform noise on specification

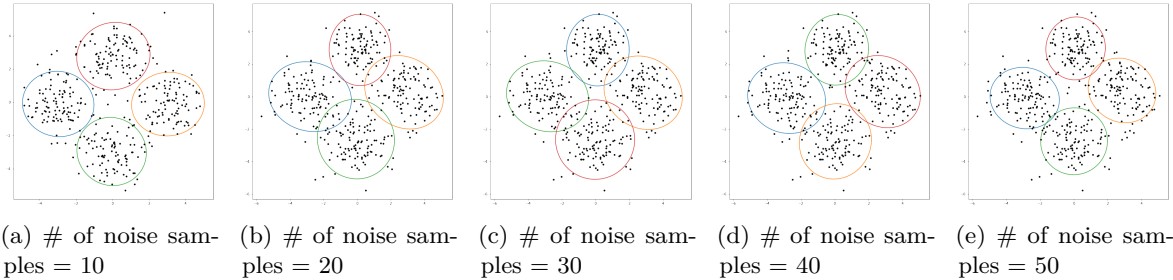

(a) # of noise samples = 10

(b) # of noise samples = 20

(c) # of noise samples = 30

(d) # of noise samples = 40

(e) # of noise samples = 50

Figure 10: SIA improves the clustering performance in the uniform noise case

Table 30: Improvement in clustering performance before and after SIA step-2 with increasing noisy samples $(10, \ldots, 50)$

| | Before SIA Step 2 | | | | | After SIA Step 2 | | | | |
|---|---|---|---|---|---|---|---|---|---|---|
| # of samples | LL | KLF | KLB | MPKL | ARI | LL | KLF | KLB | MPKL | ARI |
| 10 | -1676.9 | 75.6 | 85.2 | 4.4 | 0.91 | -1681.3 | 69.0 | 70.6 | 1.2 | 0.92 |
| 20 | -1719.6 | 85.2 | 68.2 | 8.4 | 0.87 | -1737.8 | 51.3 | 45.0 | 2.4 | 0.89 |
| 30 | -1761.1 | 64.0 | 87.2 | 6.9 | 0.88 | -1778.4 | 44.2 | 52.8 | 2.3 | 0.89 |
| 40 | -1806.2 | 78.1 | 67.3 | 5.6 | 0.87 | -1815.9 | 55.2 | 51.7 | 2.2 | 0.89 |
| 50 | -1839.6 | 79.4 | 67.1 | 6.2 | 0.87 | -1845.5 | 70.2 | 65.7 | 2.0 | 0.88 |

### F.3 Inferior Clusterings at High Dimensions: Corruption with a Student's t distribution

In this section, we show that under misspecification using Student's-$t$ corruption, the asymmetric fitted components observed in 2-dimensional settings are also present for higher dimensions. Our simulation setting is as follows: There are 3 clusters whose means are given by $(0, \ldots, 0)_p; (-3, \ldots, -3)_p; (3, \ldots, 3)_p$ and all of them have unit covariances. For each cluster, $50 \times p$ datapoints are simulated from a Gaussian distribution and $50 \times p$ datapoints are simulated from a Student's-t distribution. The degrees of freedom of the Student's-t distribution is set to 2 or 10 depending on the degree of misspecification. We fit a 3 component GMM in this misspecified setting. Since it is not possible to visualize in high dimensional settings, to capture the asymmetry in the fitted components, we compute the maximum difference between the weights of the fitted components $\Delta_w$, maximum difference between the determinants of fitted covariances $\Delta_c$, and the ARI of the clustering solution. When the misspecification is small (D.o.F $= 10$), then the values of $\Delta_c$ and $\Delta_w$ should be relatively small compared to when the misspecification is large (D.o.F $= 2$). The dimensionality $p$ to chosen from $\{2, 5, 10, 50, 100\}$. For each setting, we simulate 50 different datasets and the average results are given in table 31. We can see that increasing dimensionality (while keeping the proportion of corrupted points the same) increases the asymmetry of fitted components and the clustering performance deteriorates.

### F.4 Inferior Clusterings at High Dimensions: Contamination with uniform noise

Table 31: Inferior Clusterings at High Dimensions: Corruption with a Student's t distribution

| p | D.o.F $= 2$ | | | D.o.F $= 10$ | | |
|---|---|---|---|---|---|---|
| | $\Delta_c$ | $\Delta_w$ | ARI | $\Delta_c$ | $\Delta_w$ | ARI |
| 2 | 4e4 | 0.73 | 0.08 | 4e0 | 0.114 | 0.741 |
| 5 | 6e11 | 0.84 | 1e-4 | 8e2 | 0.497 | 0.271 |
| 10 | 8e21 | 0.83 | 0.0 | 2e5 | 0.569 | 0.024 |
| 50 | 4e101 | 0.78 | 0.0 | 8e24 | 0.656 | 0.0 |
| 100 | 1e189 | 0.77 | 0.0 | 1e47 | 0.658 | 0.0 |

In this section, we show that under misspecification using uniform noise, the asymmetric fitted components observed in 2-dimensional settings are also present for higher dimensions. Our simulation setting is as follows: There are 3 clusters whose means are given by $(0, \ldots, 0)_p; (-3, \ldots, -3)_p; (3, \ldots, 3)_p$ and all of them have unit covariances. For each cluster, $100 \times p$ datapoints are simulated from a Gaussian distribution. We add $u_n \times p$ datapoints sampled uniformly from the square $(-6, \ldots, -6)_p \times (6, \ldots, 6)_p$ as noise. The value of $u_n$ is set to 10 or 50 to control for the degree of misspecification. We fit a 3 component GMM in this misspecified setting. Since it is not possible to visualize in high dimensional settings, to capture the asymmetry in the fitted components, we compute the maximum difference between the weights of fitted components $\Delta_w$, maximum difference between determinants of fitted covariances $\Delta_c$, and the ARI of the clustering solution. When the misspecification is small ($u_n = 10$), then the values of $\Delta_c$ and $\Delta_w$ should be relatively small compared to when the misspecification is large ($u_n = 50$). The dimensionality $p$ to chosen from $\{2, 5, 10, 50, 100\}$. For each setting, we simulate 50 different datasets and the average results are given in table 32. We can see that increasing dimensionality (while keeping the proportion of corrupted points the same) decreases the asymmetry in fitted components' weights and the clustering performance increases.

Table 32: Inferior Clusterings at High Dimensions: Contamination with uniform noise

| p | $u_n = 10$ | | | $u_n = 50$ | | |
|---|---|---|---|---|---|---|
| | $\Delta_c$ | $\Delta_w$ | ARI | $\Delta_c$ | $\Delta_w$ | ARI |
| 2 | 0.56 | 0.015 | 0.93 | 2.88 | 0.038 | 0.92 |
| 5 | 0.98 | 0.004 | 0.99 | 17.65 | 0.025 | 0.99 |
| 10 | 1.15 | 0.002 | 0.99 | 42.21 | 0.014 | 0.99 |
| 50 | 1.42 | 0.00 | 1 | 104.80 | 0.003 | 1 |
| 100 | 1.19 | 0.00 | 1 | 101.25 | 0.001 | 1 |

## G  Theoretical Results

### G.1  Proof of Theorem 1

#### G.1.1  Upper bound on the KL-divergences between the true distribution and fitted distribution

Let true distribution be $G^* = \pi \mathcal{N}(-\mu, \sigma^2) + \pi \mathcal{N}(\mu, \sigma^2) + (1 - 2\pi)\mathcal{N}(\mu, b^2\sigma^2)$. WLOG, assume that $\mu > 0$. Let the fitted distribution be $G'(\boldsymbol{\theta}) = (1 - \pi')\mathcal{N}(\mu_1, \sigma_1^2) + \pi'\mathcal{N}(\mu_2, \sigma_2^2)$, where $\boldsymbol{\theta} = (\mu_1, \sigma_1^2, \mu_2, \sigma_2^2, \pi')$.

Let $\boldsymbol{\theta}^* = (\mu, \sigma^2, \mu, \sigma^2, \pi)$. Clearly, $G'(\boldsymbol{\theta}^*) \neq G^*$ . Dwivedi et al. (2018) derive the upper bound on the KL-divergence between true distribution and fitted (misspecified) distribution, when there is only one unknown parameter; we follow a similar approach.

We give a brief definition of Couplings of discrete measures. We define two discrete measures $D = \sum_k \pi_k \delta_{\eta_k}$ and $D' = \sum_{k'} \pi'_{k'} \delta_{\eta_{k'}}$, where $\delta_\eta$ is the Dirac Delta function at $\eta = (\mu, \sigma^2) \in \mathbb{R} \times \mathbb{R}^+$. The set of Couplings $\Pi(D, D')$ between two measures $D$ and $D'$ is defined as

$$\Pi(D, D') = \left\{ T \in \mathbb{R}_+^{k \times k'} : T 1_{k'} = \pi, T^\top 1_k = \pi' \right\} \tag{9}$$

where $\pi = (\pi_1, \ldots, \pi_k)^T$, $\pi' = (\pi'_1, \ldots, \pi'_{k'})^T$, and $1_k$ denotes a $k$ -dimensional vector with all entries equal to 1. In other words, $\Pi(G, G')$ is the set of all joint distributions $T$ on the space $[k] \times [k']$ such that the marginals of the distribution $T$ are equal to $\pi$ and $\pi'$ (Dwivedi et al., 2018). Couplings can be viewed as a discretized version of Copulae (Rajan & Bhattacharya, 2016; Kasa et al., 2019).

Lemma 1 of Nguyen et al. (2013) gives an upper bound on the KL-divergence of a two mixtures in terms of their individual components as follows:

$$\mathrm{KL}\left(G^*, G'(\boldsymbol{\theta})\right) \leq \inf_{\mathbf{h} \in \Pi(D, D')} \sum_{ij} h_{ij} \mathrm{KL}\left(f_i, f'_j(\boldsymbol{\theta})\right) \tag{10}$$

where $\mathbf{h}$ is a coupling, and $f_i$ and $f'_j$ are individual components of the mixtures $G$ and $G'$ respectively.

Define

$$\bar{G} := G'(\bar{\theta}) \quad \text{where} \quad \bar{\theta} \in \arg\min_{\theta \in \Theta} \mathrm{KL}\left(G_*, G'(\theta)\right) \tag{11}$$

Therefore, from equation 10, we have

$$\mathrm{KL}\left(G^*, G'(\bar{\boldsymbol{\theta}})\right) = \min_{\boldsymbol{\theta}} \mathrm{KL}\left(G^*, G'(\boldsymbol{\theta})\right) \leq \min_{\boldsymbol{\theta}} \inf_{\mathbf{h} \in \Pi(D, D')} \sum_{ij} h_{ij} \mathrm{KL}\left(f_i, f'_j(\boldsymbol{\theta})\right) \leq \inf_{\mathbf{h} \in \Pi(D, D')} \sum_{ij} h_{ij} \mathrm{KL}\left(f_i, f'_j(\boldsymbol{\theta}^*)\right) \tag{12}$$

Let $\mathbf{h} = \begin{bmatrix} \pi & 0 \\ \pi & 0 \\ 1 - \bar{\pi} - 2\pi & \bar{\pi} \end{bmatrix}$

Substituting this in the RHS of the inequality 10 and setting $\boldsymbol{\theta} = \boldsymbol{\theta}^*$, we get

$$\mathrm{KL}\left(G^*, G'(\bar{\boldsymbol{\theta}})\right) \leq (1 - 2\bar{\pi})(-\log b + 0.5b^2 - 0.5) + \bar{\pi}\frac{2\mu^2}{\sigma^2} =: C_2 \tag{13}$$

where, $C_2$ is a known constant that depends only on the parameters of the true distribution.

#### G.1.2  Lower bound on the KL-divergence between the true distribution and fitted distribution

One approach to lower bound the KL-divergence is to use the proof ideas from Pinsker's inequality (Pinsker, 1964; Lattimore & Szepesvári, 2020). In particular, we use the data processing inequality to derive a lower bound on $\mathrm{KL}\left(G^*, G'(\bar{\boldsymbol{\theta}})\right)$. Before we proceed with the proof, we need the following propositions.

**Proposition 1** (Pinsker's inequality for Bernoulli Random Variables). *Let $p, q \in [0, 1]$ be the parameters of two Bernoulli distributions $\mathbb{P}$ and $\mathbb{Q}$. Then*

$$KL\left(\mathbb{P}, \mathbb{Q}\right) \geq 2(p - q)^2. \tag{14}$$

**Proposition 2** (The BH Bound). *Let $m, m'$ be univariate distributions with support on the real line $\mathbb{R}$.*

$$\mathrm{TV}(m, m') \leq \sqrt{1 - e^{-KL(m, m')}}. \tag{15}$$

The proof of Proposition 1 can be found in Lattimore & Szepesvári (2020). Proposition 2 is a more stricter alternative to Pinsker's inequality using the Bertagonolle Huber bound (Bretagnolle & Huber, 1978) - refer to Clement Canonne's article for a more detailed review [4].

Consider the sets $\mathbb{R}^-, \mathbb{R}^+$ (i.e. the -ve and +ve real lines) in $\mathbb{R}$. Let the random variable $X$ be distributed according to $G^*$ and the random variable $Y$ be distributed according to $G'(\bar{\boldsymbol{\theta}})$. Consider the random variables $X_b = \mathrm{I}_{\mathbb{R}^-}(X)$ and $Y_b = \mathrm{I}_{\mathbb{R}^-}(Y)$, where $\mathrm{I}$ is the indicator function. Clearly, the random variable $X_b$ has a Bernoulli distribution with parameter $\frac{1}{2} + C_w$, where $C_w = \frac{1-2w}{2}\mathrm{erf}\left(\frac{-\mu}{\sqrt{2}\sigma}\right)$. The random variable $Y_b$ has a Bernoulli distribution with parameter $(1 - \bar{p})(0.5(1 + \mathrm{erf}(\frac{\bar{\mu}_1}{\sqrt{2}\bar{\sigma}_1}))) + (\bar{p})(0.5(1 + \mathrm{erf}(\frac{\bar{\mu}_2}{\sqrt{2}\bar{\sigma}_2})))$. Then by using the data processing inequality, we have

$$\mathrm{KL}\left(G^*, G'(\bar{\boldsymbol{\theta}})\right) \geq \mathrm{KL}\left(\mathcal{B}(0.5 + C_w), \mathcal{B}\left((1 - \bar{p})(0.5(1 + \mathrm{erf}(\frac{\bar{\mu}_1}{\sqrt{2}\bar{\sigma}_1}))) + (\bar{p})(0.5(1 + \mathrm{erf}(\frac{\bar{\mu}_2}{\sqrt{2}\bar{\sigma}_2})))\right)\right) \tag{16}$$

Next using proposition 1, we can write

$$\mathrm{KL}\left(G^*, G'(\bar{\boldsymbol{\theta}})\right) \geq 2\left(C_w - 0.5(\bar{p}\,\mathrm{erf}(\frac{\bar{\mu}_2}{\sqrt{2}\bar{\sigma}_2}) + (1 - \bar{p})\mathrm{erf}(\frac{\bar{\mu}_1}{\sqrt{2}\bar{\sigma}_1}))\right)^2 \tag{17}$$

Combining this lower bound with the upper bound obtained in equation 13, we have

$$2\left(-\sqrt{\frac{C_2}{2}} + C_w\right) \leq \tau(\bar{\theta}) \leq 2\left(\sqrt{\frac{C_2}{2}} + C_w\right) \tag{18}$$

where, $\tau(\bar{\theta}) := \left((\bar{p})\mathrm{erf}(\frac{\bar{\mu}_2}{\sqrt{2}\bar{\sigma}_2}) + (1 - \bar{p})\mathrm{erf}(\frac{\bar{\mu}_1}{\sqrt{2}\bar{\sigma}_1})\right)$

Alternately, we can also use proposition 2 by noting that total variation distance of two Bernoulli distributions $\mathbb{P}, \mathbb{Q}$ with parameters $p, q$ is $|p - q|$, and hence proposition 2 leads to the following inequality:

$$-\log(1 - (p - q)^2) \leq \mathrm{KL}(\mathbb{P}, \mathbb{Q}). \tag{19}$$

Applying this inequality in (16) and (13), we have

$$C_2 \geq \mathrm{KL}\left(G^*, G'(\bar{\boldsymbol{\theta}})\right) \geq -\log\left(1 - \left(\frac{\tau(\bar{\theta})}{2} - C_w\right)^2\right) \tag{20}$$

---

[4] https://github.com/ccanonne/probabilitydistributiontoolbox/blob/master/pinskers-and-beyond.pdf

This can be rewritten as

$$2\left(-\sqrt{1-e^{-C_2}}+C_w\right) \le \tau(\bar{\theta}) \le 2\left(\sqrt{1-e^{-C_2}}+C_w\right) \tag{21}$$

Next we empirically compare the bounds obtained using approach 1 and approach 2 for different $(\mu, \sigma, \pi)$ for varying values of $t$. For a given value of $t$, we simulate 200 datasets and pick the maximum of $\tau(\bar{\theta}) = \left|\left(\bar{p}\mathrm{erf}(\frac{\bar{\mu}_2}{\sqrt{2}\bar{\sigma}_2}) + (1-\bar{p})\mathrm{erf}(\frac{\bar{\mu}_1}{\sqrt{2}\bar{\sigma}_1})\right)\right|$ among these 200 output solutions. From figure 11, we see that the bound is most effective when the separation of the components in the true distribution is low and proportion of the contaminating distribution is high. Also, we note that approach -1 gives a tighter bound for most values of $b$ and hence we go with it in our theorem 1.

It is possible to generalise Theorem 1 to the multivariate case. Here we have considered this particular univariate setup because of the availability of closed form expressions of KL-divergences. For general settings, using this same approach, it is possible to compute similar bounds if the closed form expressions for KL-divergences are available.

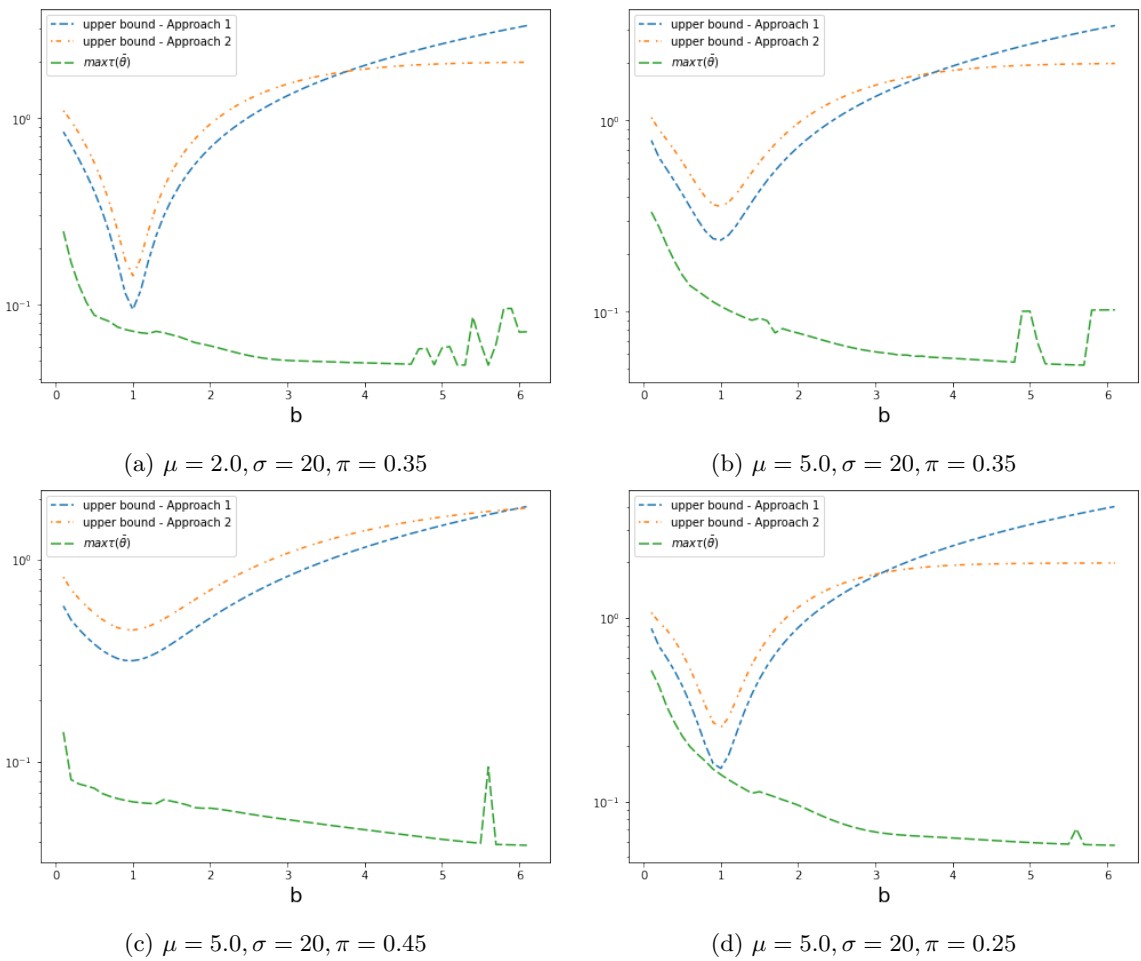

Figure 11: Empirical comparison of the bounds for different $(\mu, \sigma, \pi)$ for varying values of $b$

## G.2 Proof of Theorem 2

First, we show the boundedness for the simple case of a univariate mixture model as it is straightforward to follow and later proceed to the proof of the multivariate case.

*Proof.* **Univariate:** With loss of generality, let the component 1 - $\mathcal{N}(x; \mu_1, \sigma_1^2)$ - is the one that is fitting over a single point and reducing its variance $\hat{\sigma}_1^2 \to 0$; also, assume that $\hat{\sigma}_1^2 \le \hat{\sigma}_2^2 \le c < \infty$

First note that the value of likelihood satisfies the following inequality:

$$\mathcal{N}(x; \mu, \sigma^2) \le \frac{1}{\sqrt{2\pi}\sigma} \tag{22}$$

For one-dimensional case, the penalized likelihood of SIA for an observation $x_i$ can be written as follows:

$$\log(\hat{\pi}_1 \mathcal{N}(x_i; \hat{\mu}_1, \hat{\sigma}_1^2) + \hat{\pi}_2 \mathcal{N}(x_i; \hat{\mu}_2, \hat{\sigma}_2^2)) - w_1 \left( \log \frac{\hat{\sigma}_2}{\hat{\sigma}_1} + \frac{\hat{\sigma}_1^2 + (\hat{\mu}_1 - \hat{\mu}_2)^2}{2\hat{\sigma}_2^2} - \frac{1}{2} \right) - w_2 \left( \log \frac{\hat{\sigma}_1}{\hat{\sigma}_2} + \frac{\hat{\sigma}_2^2 + (\hat{\mu}_1 - \hat{\mu}_2)^2}{2\hat{\sigma}_1^2} - \frac{1}{2} \right) \tag{23}$$

(By using eq 22 and setting $(w_3 = w_2(\hat{\sigma}_2^2 + (\hat{\mu}_1 - \hat{\mu}_2)^2))$, and leaving out -ve terms)

$$\le \log(\frac{\hat{\pi}_1}{\hat{\sigma}_1} + \frac{\hat{\pi}_2}{\hat{\sigma}_2}) - w_2 \log \hat{\sigma}_1 - \frac{w_3}{2\hat{\sigma}_1^2} - \log \sqrt{2\pi} + w_2 \log c \tag{24}$$

$$\le \log(\frac{\hat{\pi}_1 + \hat{\pi}_2}{\hat{\sigma}_1}) - w_2 \log \hat{\sigma}_1 - \frac{w_3}{2\hat{\sigma}_1^2} + w_2 \log c (\text{where } w_4 = 1 + w_2)$$

$$\le -w_4 \log \hat{\sigma}_1 - \frac{w_3}{2\hat{\sigma}_1^2} + w_2 \log c \tag{25}$$

(Differentiating wrt $\sigma_1$ and setting to 0, we find $\sigma_1^* = \sqrt{\frac{w_3}{w_4}}$)

$$\le -w_4 \log \sqrt{\frac{w_3}{w_4}} - 0.5w_4 + w_2 \log c \tag{26}$$

The terms in equation 26 are all constants hence the SIA likelihood is bounded. It should be noted that since we started with the assumption that only the first component becomes degenerate (i.e. $\sigma_1^2 \to 0$) while the second component doesn't, the same boundedness result can be trivially generalized to the case where there are more than 2 components.

**Multivariate:** $\lambda_1(\hat{\Sigma}_1), \lambda_1(\hat{\Sigma}_2) \le c_1$ where $\lambda_p(.)$ and $\lambda_1(.)$ denotes the smallest and the largest eigenvalue of the $\mathbb{R}^{p \times p}$ matrix. The likelihood can be made unbounded by making the determinant of $\hat{\Sigma}_1$ close to zero i.e. $|\hat{\Sigma}_1| \to 0$.

First, we bound the following terms:

$$0.5w_2 \log(|\hat{\Sigma}_2|) \le 0.5w_2 p \log(c_1) = C(\text{some constant}) \tag{27}$$

$$\log(\hat{\pi}_1 \mathcal{N}(x_i; \hat{\mu}_1, \hat{\Sigma}_1) + \hat{\pi}_2 \mathcal{N}(x_i; \hat{\mu}_2, \hat{\Sigma}_2)) - 0.5 * w_1 \left( \log \frac{|\hat{\Sigma}_2|}{|\hat{\Sigma}_1|} + \text{tr}(\hat{\Sigma}_2^{-1} \hat{\Sigma}_1) + (\mu_2 - \mu_1)^T \hat{\Sigma}_2^{-1} (\mu_2 - \mu_1)^T \right)$$

$$- 0.5 * w_2 \left( \log \frac{|\hat{\Sigma}_1|}{|\hat{\Sigma}_2|} + \text{tr}(\hat{\Sigma}_1^{-1} \hat{\Sigma}_2) + (\mu_2 - \mu_1)^T \hat{\Sigma}_1^{-1} (\mu_2 - \mu_1)^T \right) \tag{28}$$

(By using eq 22 and 27, and leaving out -ve terms and absorbing the constants into $C$)

$$\leq 0.5 * \log(\frac{\hat{\pi}_1}{|\hat{\Sigma}_1|} + \frac{\hat{\pi}_2}{|\hat{\Sigma}_2|}) - 0.5 * w_2 \log(|\hat{\Sigma}_1|) - 0.5 * w_2 \text{tr}(\hat{\Sigma}_1^{-1} \hat{\Sigma}_2) \tag{29}$$

$$= 0.5 * \left( \log(\frac{\hat{\pi}_1 + \hat{\pi}_2}{|\hat{\Sigma}_1|}) - w_2 \log(|\hat{\Sigma}_1|) - w_2 \text{tr}(\hat{\Sigma}_1^{-1} \hat{\Sigma}_2) \right) \tag{30}$$

$$= 0.5 * \left( - \log(|\hat{\Sigma}_1|) - w_2 \log(|\hat{\Sigma}_1|) - w_2 \text{tr}(\hat{\Sigma}_1^{-1} \hat{\Sigma}_2) \right) \tag{31}$$

$$= 0.5 * \left( -w_3 \log(|\hat{\Sigma}_1|) - w_2 \text{tr}(\hat{\Sigma}_1^{-1} \hat{\Sigma}_2) \right) \quad (\text{where } w_3 = 1 + w_2) \tag{32}$$

$$= 0.5 * \left( w_3 \log(|\hat{\Sigma}_1^{-1}|) - w_2 \text{tr}(\hat{\Sigma}_1^{-1} \hat{\Sigma}_2) \right) \tag{33}$$

Note that equation 33 is concave in $\Sigma_1^{-1}$ as $\log(|.|)$ is a concave function and trace is affine (Boyd & Vandenberghe, 2004); hence, if a maximum exists, it is the global maximum. We can find the maximum by differentiating with respect to $\Sigma_1^{-1}$ and setting to zero;

$$\nabla_{\Sigma_1^{-1}} \left( w_3 \log(|\hat{\Sigma}_1^{-1}|) - w_2 \text{tr}(\hat{\Sigma}_1^{-1} \hat{\Sigma}_2) \right) = w_3 \hat{\Sigma}_1 - w_2 \hat{\Sigma}_2 = 0 \tag{34}$$

$$\implies \hat{\Sigma}_1 = \frac{w_2}{w_3} \hat{\Sigma}_2 \tag{35}$$

Therefore, the final upper bound of the log-likelihood is given by

$$-0.5(1 + w_2) \left( \log(|\frac{w_2}{1 + w_2} \hat{\Sigma}_2|) + p \right) + C \tag{36}$$

$\square$

## H    Closed Forms

In this section, we illustrate the difficulty of deriving the closed form expressions of the gradients of the penalized likelihood used in SIA. These closed form expressions are required to develop an EM algorithm for inference. Consider the mean update in EM for the unpenalized loglikelihood:

$$\frac{\partial \log \mathcal{L}}{\partial \boldsymbol{\mu}_k} = -\sum_{i=1}^{n} \gamma(z_{ik}) \boldsymbol{\Sigma}_k^{-1} (\mathbf{x}_i - \boldsymbol{\mu}_k) \tag{37}$$

where $\gamma(z_{ik})$ is defined as the responsibility and is equal to $\frac{\pi_k \mathcal{N}(\mathbf{x}_i | \boldsymbol{\mu}_k, \boldsymbol{\Sigma}_k)}{\sum_{j=1}^{K} \pi_j \mathcal{N}(\mathbf{x}_i | \boldsymbol{\mu}_j, \boldsymbol{\Sigma}_j)}$. Setting $\frac{\partial \log \mathcal{L}}{\partial \boldsymbol{\mu}_k}$ to zero, we obtain $\boldsymbol{\mu}_k = \frac{\sum_{i=1}^{n} \gamma(z_{ik}) \mathbf{x}_i}{\sum_{i=1}^{n} \gamma(z_{ik})}$. Note that the computation of the mean update $\boldsymbol{\mu}_k$ can be easily parallelized in the M-step because it does not involve other mean terms $\boldsymbol{\mu}_j$ where $j \neq k$.

Now, consider the gradient of the penalized loglikelihood:

$$\frac{\partial \log(\mathcal{L} - w_1 \times KLF - w_2 \times KLB)}{\partial \boldsymbol{\mu}_k} = -\sum_{i=1}^{n} \gamma(z_{ik}) \boldsymbol{\Sigma}_k^{-1} (\mathbf{x}_i - \boldsymbol{\mu}_k) - w_1 \sum_{j \neq k} 2\boldsymbol{\Sigma}_j^{-1} (\boldsymbol{\mu}_k - \boldsymbol{\mu}_j) - w_2 \sum_{j \neq k} 2\boldsymbol{\Sigma}_k^{-1} (\boldsymbol{\mu}_k - \boldsymbol{\mu}_j)$$
$$\tag{38}$$

Setting the above expression to zero, we can obtain the expression for the mean $\boldsymbol{\mu}_k$ as:

$$
\begin{aligned}
\boldsymbol{\mu}_k = & \left( \sum_{i=1}^{n} \gamma(z_{ik}) \boldsymbol{\Sigma}_k^{-1} - (w_1 + w_2)(\sum_{j \neq k} \left( \boldsymbol{\Sigma}_j^{-1} + \boldsymbol{\Sigma}_k^{-1} \right)) \right)^{-1} \\
& \left( \sum_{i=1}^{n} \gamma(z_{ik}) \boldsymbol{\Sigma}_k^{-1} \mathbf{x}_i - (w_1 + w_2)(\sum_{j \neq k} \left( \boldsymbol{\Sigma}_j^{-1} + \boldsymbol{\Sigma}_k^{-1} \right) \boldsymbol{\mu}_j) \right)
\end{aligned}
\tag{39}
$$

As seen above, even though the process of obtaining the updates is straightforward, it is laborious. For the update of covariance estimates, we could not find a closed form estimate for the penalized loglikelihood. Further, note the mean update requires knowledge of other mean terms $\boldsymbol{\mu}_j$ and hence cannot be parallelized. EM based updates for the more flexible models such as MFA will be even more cumbersome.

In contrast, using Gradient Descent (GD) based inference using Automatic Differentiation (AD) avoids the need to compute such laborious gradient updates by hand or to hard code them in the software program. AD-GD based inference approach is model-agnostic as AD software are blackbox tools where we just need to input the loglikelihood; they don't require closed forms of the gradients. Thus the same software implementation can be easily extended for more flexible models such as MFA, PGMM etc. This is not the case with the traditional EM based inference where the updates have to be derived and re-implemented for each model.

## I  Likelihood Surface Visualization

To understand how the penalization in (1) helps in clustering we visualize the log-likelihood surface using the technique of Li et al. (2018) for neural network loss landscapes. Given two sets of parameters $\boldsymbol{\theta}_1, \boldsymbol{\theta}_2$, the surface function $S(\alpha, \beta) = \mathcal{L}(\alpha\boldsymbol{\theta}_1 + \beta\boldsymbol{\theta}_2)$ where $0 \le \alpha, \beta \le 1$ is used to analyze the log-likelihood $\mathcal{L}$.

As an illustration, we use the Pinwheel data described in section 3 with the parameters obtained for (g) and (a) of fig. 1 as $\boldsymbol{\theta}_1$ and $\boldsymbol{\theta}_2$ respectively and plot the log-likelihood and penalized log-likelihood in fig. 12 (a) and (b) respectively. It is evident in fig. 12 (a) that the log-likelihood has a significant plateau region which has almost similar (unpenalized) likelihood. However, for SIA likelihood in (b), such a plateau does not exist and there exists a clear maxima at $(\alpha = 1, \beta = 0)$. As expected $\boldsymbol{\theta}_1$ ((g) of fig. 1) has a better clustering output, despite having similar (unpenalized) likelihood as that of $\boldsymbol{\theta}_2$.

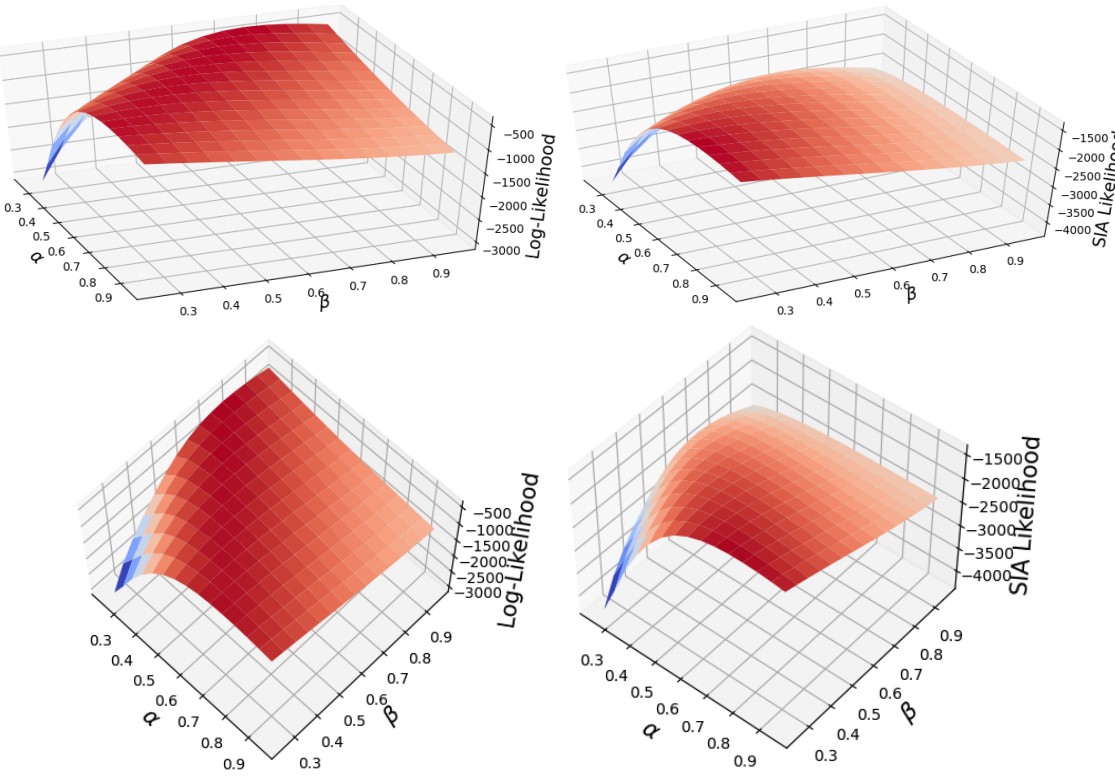

Figure 12: (Left:) Side-view and Top-view of GMM Log-likelihood show a plateau region ; (Right:) Side-view and Top-view of Penalized log-likelihood has no plateau and a clear peak corresponding to good clustering.

## J  Wallclock time

We benchmark the runtime of SIA with respect to AD-GD and EM. We do so by simulating 50 datasets, each containing 100 datapoints sampled randomly, for each setting. We run all the algorithms to convergence in likelihood (tolerance 1e-5) or a maximum of 50 iterations, which ever happens earlier. We evaluate with the number of clusters $K = \{3, 5\}$ and data dimensions $p = \{2, 5, 10, 50\}$. All experiments were run using Autograd, Numpy and Sklearn packages in Python 3.7 on Dell Windows 10 machine (Intel i7-6700 quadcore CPU@3.40GHz; 8 GB RAM; 500 GB HDD).

Table 33 shows the runtime of all the algorithms. We observed that with increasing dimensionality, the number of iterations for EM reduced, and at $p = 50$, EM failed to go beyond iteration 1. As expected, SIA takes roughly double the time taken by AD-GD.

Table 33: Average Runtime in seconds with data dimensions $2, 5, 10, 50$.

| # of clusters | Algorithm | 2 | 5 | 10 | 50 |
|---|---|---|---|---|---|
| 3 | SIA | 7.900 | 7.926 | 8.037 | 11.508 |
| 3 | AD-GD | 4.471 | 4.467 | 4.529 | 6.50 |
| 3 | EM | 0.009 | 0.006 | 0.009 | - |
| 5 | SIA | 20.571 | 20.785 | 21.238 | 30.041 |
| 5 | AD-GD | 11.127 | 11.194 | 11.433 | 16.167 |
| 5 | EM | 0.036 | 0.025 | 0.014 | - |

## K   Illustration of unbalanced clusters on Pinwheel data

We illustrate the effect of unbalanced clusters on the same pinwheel data that is discussed in section 3. In the first experiment, instead of (100,100,100) number of points respectively in the three clusters, we have (100,50,20) number of points respectively for the three clusters. Please refer to fig. 13 (a) and (b) for the clustering results with SIA with this dataset. In the second experiment we have (100,50,50) number of points. Please refer to fig. 13 (c) and (d) for the results obtained from SIA.

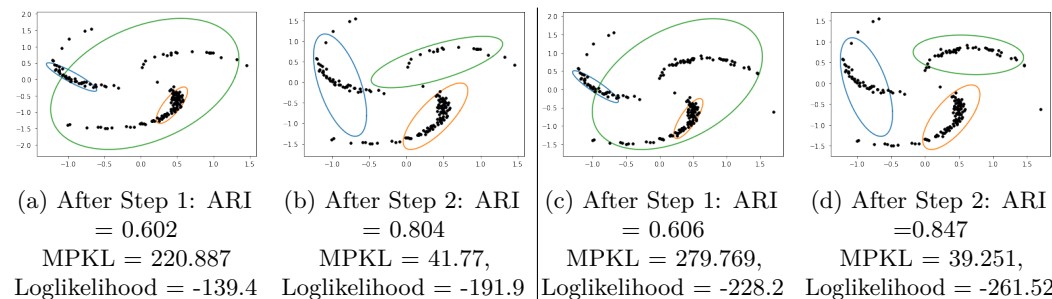

(a) After Step 1: ARI = 0.602 MPKL = 220.887 Loglikelihood = -139.4   (b) After Step 2: ARI = 0.804 MPKL = 41.77, Loglikelihood = -191.9   (c) After Step 1: ARI = 0.606 MPKL = 279.769, Loglikelihood = -228.2   (d) After Step 2: ARI =0.847 MPKL = 39.251, Loglikelihood = -261.52

Figure 13: Clustering using SIA for unbalanced pinwheel data. Componentwise number of datapoints: $\{100, 50, 20\}$ (a and b), $\{100, 50, 50\}$ (c and d).

## L   Additional Experiments

### L.1   Performance of SIA

We evaluate the performance of SIA in cases of under-specification and over-specification of number of components. For these experiments, introducing misspecification by cubing the data points (as done in simulations in section 5) lead to mixing of the components and does not allow us to study the effects of over–/under-specification of components. So, we introduce misspecification through contamination by t-distribution as described below.

### L.1.1   Under-specified number of components

We sample 120 datapoints from 3-component $p$-dimensional GMMs (40 datapoints from each component) with spherical covariances. The means of the components are $(0, \ldots, 0)_p$, $(-1, \ldots, -1)_p$ and $(1, \ldots, 1)_p$. In addition, we sample 30 datapoints from 3 component $p$-dimensional t-distribution mixture (10 datapoints from each component), with the same means and covariances, with two degrees of freedom. Thus, there are 150 datapoints sampled from a contaminated GMM. The dimensionality of $p$ is chosen from $\{2, 3, 5, 7, 10\}$. We fit a *misspecified* 2-component GMM over these 150 datapoints. For each setting, we simulated 50 datasets and compute ARI of the clustering output. The results are given in Table 34. We find that both EM and MClust do not perform well when compared to AD-GD and SIA.

### L.1.2 Over-specified number of components

We sample 80 datapoints from 2 component $p$-dimensional GMMs (40 datapoints from each component) with spherical covariances. The means of the components are $(0, \ldots, 0)_p$ and $(1, \ldots, 1)_p$. In addition, we sample 20 datapoints from 2 component $p$-dimensional t-distribution mixture, with the same means and covariances, with three degrees of freedom. Thus, the overall 100 datapoints are sampled from a contaminated GMM. The dimensionality of $p$ is chosen from $\{2, 3, 5, 7, 10\}$. We fit a *misspecified* 3-component GMM over these 100 datapoints. For each setting, we simulated 50 datasets and compute ARI of the clustering output. The results are given in Table 35.

Table 34: Average (std. dev) ARI with increasing dimensionality ($p$) for under-specified mixtures

| $p$: | 2 | 3 | 5 | 7 | 10 |
|---|---|---|---|---|---|
| SIA | **0.240** | **0.279** | **0.330** | **0.306** | **0.328** |
| | **(0.127)** | **(0.145)** | **(0.158)** | **(0.191)** | **(0.196)** |
| AD-GD | 0.238 | 0.277 | 0.323 | 0.302 | 0.323 |
| | (0.126) | (0.144) | (0.155) | (0.189) | (0.193) |
| EM | 0.042 | 0.021 | 0.066 | 0.047 | 0.100 |
| | (0.103) | (0.074) | (0.118) | (0.115) | (0.167) |
| MClust | 0.055 | 0.046 | 0.1289 | 0.092 | 0.125 |
| | (0.121) | (0.114) | (0.186) | (0.175) | (0.200) |

Table 35: Average (std. dev) ARI with increasing dimensionality ($p$) for over-specified mixtures

| $p$: | 2 | 3 | 5 | 7 | 10 |
|---|---|---|---|---|---|
| SIA | **0.189** | **0.266** | 0.369 | 0.517 | 0.614 |
| | **(0.090)** | **(0.124)** | (0.166) | (0.182) | (0.205) |
| AD-GD | 0.182 | 0.252 | 0.356 | 0.501 | 0.607 |
| | (0.089) | (0.122) | (0.160) | (0.179) | (0.203) |
| EM | 0.059 | 0.081 | 0.129 | 0.147 | 0.359 |
| | (0.094) | (0.096) | (0.134) | (0.181) | (0.244) |
| MClust | 0.074 | 0.168 | **0.381** | **0.523** | **0.666** |
| | (0.094) | (0.127) | **(0.132)** | **(0.137)** | **(0.120)** |

### L.2 Model Selection

In this section we compare the performance of MPKL with that of AIC and BIC, as model selection criteria, on synthetic data. Note that MPKL is a general model selection criterion and can be used with any inference technique and even when misspecification is not suspected.

We follow the setting used in Pan & Shen (2007); Guo et al. (2010) and simulate a 4 component 5-dimensional datasets as shown in Table 36. We assume unit spherical covariance matrices for each component. The parameters of each component across the 5 dimensions are varied as shown in Table 36 – all the components have the same parameters in 2 dimensions (4 and 5), only dimensions 1, 2, 3 are discriminating for components 2,3,4 respectively. The value of $\lambda$ controls the cluster separation. For each component we sampled ten datapoints. Three different sets of simulated data with varying cluster separation are obtained by choosing the value of $\lambda$ to be 1, 5 and 10 respectively. This experiment is repeated with 10 different seeds for each value of $\lambda$. We use SIA for clustering with each value of $K \in \{3, 4, 5\}$. We compare the number of clusters selected using AIC, BIC and MPKL.

Table 36: Simulations for Model Selection for 5-dimensional datasets. Along each dimension, each component has unit variance. Means are given below:

| Features | C-1 | C-2 | C-3 | C-4 |
|---|---|---|---|---|
| 1 | 0 | $\lambda$ | 0 | 0 |
| 2 | 0 | 0 | $\lambda$ | 0 |
| 3 | 0 | 0 | 0 | $\lambda$ |
| 4 | 0 | 0 | 0 | 0 |
| 5 | 0 | 0 | 0 | 0 |

Table 37: Number of times (in 10 reps) 3, 4, 5 components are selected with each criterion – MPKL (M), AIC (A), BIC (B) – for a 4-component 5-dimensional dataset with 3 cluster separations ($\lambda = 1, 5, 10$).

| $\lambda$: | 1 | | | 5 | | | 10 | | |
|---|---|---|---|---|---|---|---|---|---|
| #clusters | M | A | B | M | A | B | M | A | B |
| 3 | 0 | 7 | 10 | 0 | 0 | 0 | 0 | 0 | 0 |
| 4 | 10 | 3 | 0 | 10 | 10 | 10 | 10 | 10 | 10 |
| 5 | 0 | 0 | 0 | 0 | 0 | 0 | 0 | 0 | 0 |

The results are given in Table 37. When the dimensionality is low and cluster separation is small, both AIC and BIC underestimate the number of components. MPKL correctly estimates the number of components in all the cases.

We repeat the above experiment for $p = 50$; details of the experiments are given in Table 38. The parameters of each component across the 50 dimensions are varied as shown in table 38, i.e., all the components have the

same parameters for 35 dimensions, only dimensions 1-5, 5-10, 10-15 are discriminating for components 2,3,4 respectively. All other simulation settings are same as described above for the 5-dimensional case.

Table 38: Simulations for Model Selection for 50-dimensional datasets. Along each dimension, each component has unit variance. Means are given below:

| Features | C-1 | C-2 | C-3 | C-4 |
|----------|-----|-----|-----|-----|
| 1-5 | 0 | $\lambda$ | 0 | 0 |
| 5-10 | 0 | 0 | $\lambda$ | 0 |
| 10-15 | 0 | 0 | 0 | $\lambda$ |
| 15-50 | 0 | 0 | 0 | 0 |

Table 39: Number of times (in 10 reps) 3, 4, 5 components are selected with each criterion – MPKL (M), AIC (A), BIC (B) – for a 4-component 50-dimensional dataset with 3 cluster separations ($\lambda = 1, 5, 10$).

| $\lambda$: | 1 | | | 5 | | | 10 | | |
|------------|---|---|---|---|---|---|----|---|---|
| #clusters | M | A | B | M | A | B | M | A | B |
| 3 | 10 | 10 | 10 | 1 | 10 | 10 | 1 | 0 | 10 |
| 4 | 0 | 0 | 0 | 7 | 0 | 0 | 8 | 10 | 0 |
| 5 | 0 | 0 | 0 | 2 | 0 | 0 | 1 | 0 | 0 |

Table 39 shows that at very low cluster separation ($\lambda = 1$), both the criteria do not select 4 clusters. At moderate and high cluster separation ($\lambda = 5, 10$), BIC underestimates the number of clusters to 3, which is consistent with previous findings (Melnykov et al., 2010). AIC, on the other hand, performs well only when the cluster separation is very high. MPKL identifies 4 clusters in 7 out of 10 times at moderate separation and 8 out of 10 times at high cluster separation.

## M  Additional details of Wine Case Study

We give the scatterplot matrix for the recommended model S9 here in figure 14.

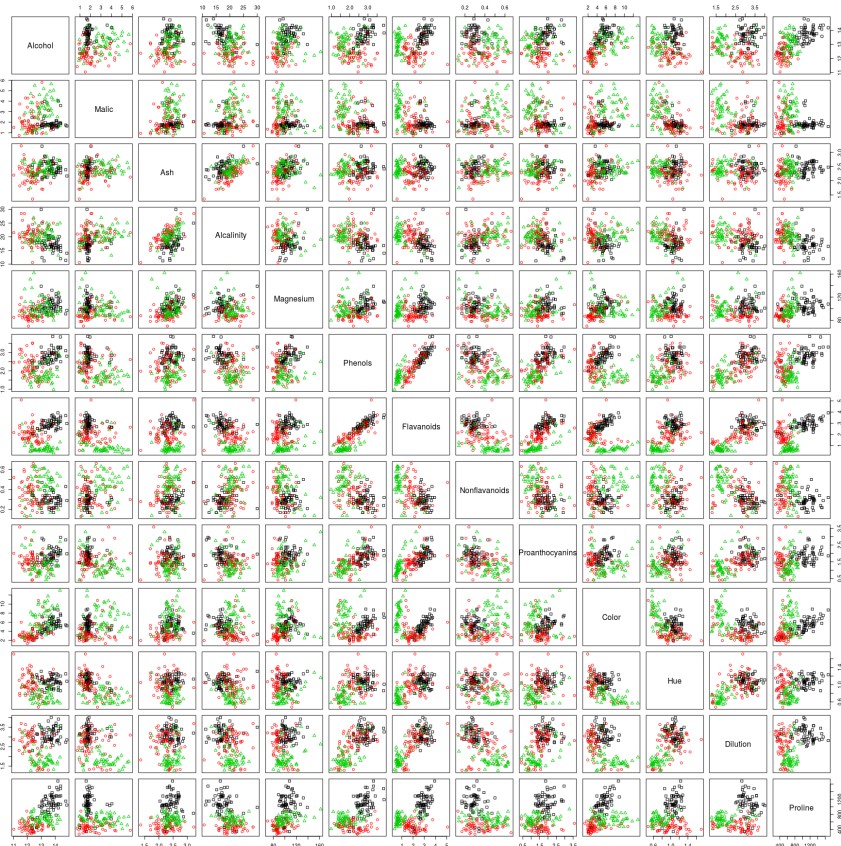

Figure 14: Scatterplot matrix for the recommended model



Figure 15: tSNE plots for the best model for each value of $K$ (S. Nos. 3,9 and 13 in Table 12); Clusters obtained for $K = 2$ (above), $K = 3$ (middle) and $K = 4$ (below).

## N   Generation of Covariance matrices

Consider a covariances matrix $\boldsymbol{\Sigma} = \boldsymbol{\Sigma}^{1/2}(\boldsymbol{\Sigma}^{1/2})^T$. The parameters of $\boldsymbol{\Sigma}^{1/2}$ are sampled randomly from a standard normal distribution to capture different covariance structures in $\boldsymbol{\Sigma}$. We illustrate that this random sampling gives different covariance structures by plotting 1000 datapoints sampled from $\mathcal{N}((0,0), \boldsymbol{\Sigma})$ for the first 10 seeds as shown in table 16.

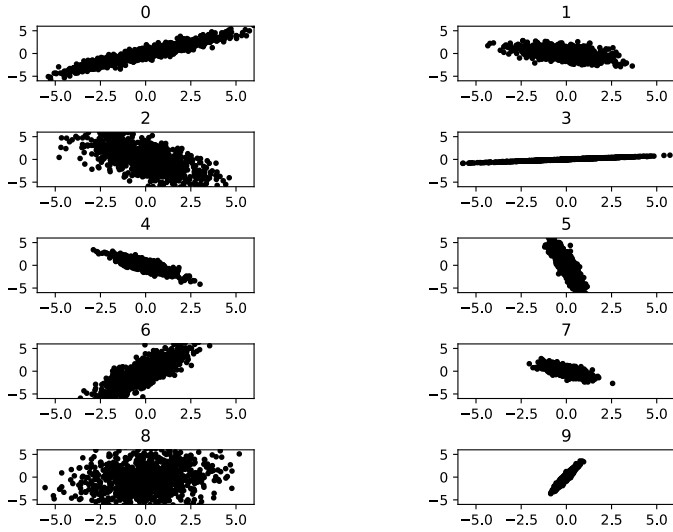

Figure 16: Different seeds give different covariances structures (seed value is mentioned above each subplot).

