# OpenReview forum: "Avoiding Inferior Clusterings with Misspecified Gaussian Mixture Models"
_TMLR — Rejected by TMLR_

### Review · Reviewer_B5oD · 2022-07-29

**Summary Of Contributions:**

Via numerical examinations (with synthetic data, in small number of dimensions), the authors define a property for the resulting solution of a GMM clustering algorithm, namely the propery of being an “inferior clustering”. Distinctions with the property of being an `spurious clustering' have been discussed. In addition, authors propose adding a KL-based term to the log likelihood (and optimization via gradient descent utilizing automatic differentiation) to discourage such ‘inferior’ clustering solutions. They also propose an algorithm, SIA, for adjusting the regularization parameters sequentially. Several numerical experiments (synthetic low dimensional data) have been provided to compare the solutions obtained by SIA with those obtained by a few other methods (possibly with a different estimation objective) in terms of a few clustering measures.


**Requested Changes:**

To improve readability, it would be helpful if the authors decoupled their observations/results as well as contribution statements regarding 1) SIA and AD, 2) defining the phenomenon of 'inferior clustering'. For example, Section 5 seems to be on SIA's superiority, while Section 6 is both on SIA's superiority and on the quality of clusters. Sections 4.2, 4.4, and 4.5 are again algorithmic, while Sections 3, 4.1, 4.3, and 4.6 focus on the clustering task itself.


I would like to know more about how the authors think a claim such as 'previously unreported class of \dots solutions'  can be justified (obviously, it is a 'non-existence' claim, hence no direct references). As an example, can the authors provide references where such clustering solutions have been categorized under other 'non-inferior' classes (i.e., solutions with an acceptable quality, where the quality in such previous work, for example, could have been defined via the value of a certain metric).


Please clarify what the following, in Section 2, means: '\dots and the fitted model has the same functional form P(.).'


Further discussions on Fig 2(b) and Fig 11 would be helpful, especially on the diverging behavior of the bound.


Could you comment on possible origins of SIA? For example, it resembles "iteratively reweighted" algorithms. Is there a connection, possibly recorded before? If so, IRLS is known to perform sparsity regularization in some data scenarios. Can a similar effect be established here, and what would that be (in stead of `sparsity promoting')?


At a basic level, what about using other notions of distance between two or more Gaussian distribution? For example, what about a simple matrix norm of the differences in covariance/precision matrices (also incorporating the means in a similar simple way)? I understand that KL provides a better comparison, but I am asking whether we know that some of the observations crucially rely on KL/MPKL or they can be derived via a much simpler notion of similarity. And in examining this question, I would like to learn more (beyond the discussions at the end of page 4) about why the authors believe the current scope of numerical investigations of this phenomenon, i.e.,
- with low-dimensional synthetic data, e.g., as described in Table 2,
- identification of the phenomenon either visually (e.g., Fig 1, 3), or, based on 'function/metric values' (only the values of the penalties, rather than say optimality conditions to connect the loss and the penalty) (e.g., Fig 1, Sec 5.1, 5.2, 6), [the use of metric values to judge clustering performance is common, but my point is in the context of 'defining' inferior clusterings as mentions in the sequel]
- an ad-hoc (as desribed in Sections 3.2 and 4.1) estimation procedure, namely the use of MPKL,

is sufficient for 'defining' a class of clustering solutions? With 'definition' here, I mean describing the in-class objects, and, distinguishing them from out-of-class objects, in a way that can be re-implemented elsewhere.


In Section 4.4, in taking the minimum MPKL, the text suggests considering all subsets of the form {1,..,K} for K in {2,..,L}. I am wondering if all subsets of {1,..,L} of size at least two should be considered instead?


In Section 7, in '\dots that, by design, avoids inferior solutions', using 'discourage' could be more accurate than 'avoid'.

I would appreciate responses to all of the above.

**Strengths And Weaknesses:**

The manuscript is well-written. It could use further references for some of the theoretical facts and observations; e.g.,
- references missing for KLF, KLB, etc, and the facts/observations mentioned about them;
- the last line of Section 4.3 on how SIA avoids degenracy based on the existence of a bound.
- Section 4.5, 'this is comparable to most GMM inference methods'
- Section 4.3, 'which is easy to verify'

I believe the simulation study, in its claimed scope, has been careful and thorough.

Appendices are helpful.

---

### Review · Reviewer_YoVD · 2022-08-03

**Summary Of Contributions:**

The authors propose an alternative objective for fitting Gaussian Mixture Models (GMMs) based on a penalised log-likelihood in which the penalty term discourages large Kullbach-Leibler (KL) divergences between the different components. Among other benefits this is a bounded objective (unlike the log-likelihood in some cases) which means that degenerate solutions cannot arise. The proposal is motivated by avoiding so-called "inferior solutions": solutions which arise in misspecified models which have a high log-likelihood (and so are favoured by methods based only on maximising likelihood (ML)) but which correspond with poor clustering performance of the mixture components.

The authors propose a two-step algorithm in which initially a ML solution, which is then used to initialise the optimisation of their penalised likelihood objective. Practically the authors use automatic differentiation in order to optimise their objective using gradient ascent, which shows good performance empirically in their experimental results.

In addition, the authors propose a new model selection criterion for GMMs based on the largest absolute difference between the KL divergences of pairs of components from one another.



**Requested Changes:**

Crucial (but not necessarily sufficient):
- Provide at least a heuristic discussion which justifies MPKL as a model selection criterion.
- If the theoretical contributions cannot be enhanced, then the experimental settings need to be vastly more extensive than they are. In particular:
-- Equal and spherical covariances are extremely restrictive and should not constitute so large a proportion of the experiments in order to justify the relevance of the method.
-- Include more than just two components when covariances are varied.
-- Include MPKL results for more of the real data sets.


Non-crucial:
- Clarify preciseness of what is meant by spurious solutions in the main paper

Queries/corrections:
- What does it mean to "cube" data?
- Sometimes G^* and sometimes G_* is used in Section 3.2.
- pg. 5: "whose values LIE in (-1, 1)..."
- When referring to a named section.table.figure, use captials, e.g. "in Section 1 we..." and not "in section 1 we..."
- What is the parameter w in Theorem 1?
- I do not think it is correct to refer to EM, Mclust, etc. as inference methods. These are estimation methods.
- Real data set IV doesn't satisfy n < p as stated.
- Proof read and anonymise your appendices as well!
- In the case study you mention that Mclust is used as well as ML methods. What form of constraint is used? Also, the results are not given for Mclust.
- In Table 12 the MPKL minimiser is solution 2 and not solution 9, so why is 9 highlighted?
- The phrase "asymmetric covariances" sounds like individual covariance matrices are asymmetric!
- pg. 3: "... number of dimensions IS greater than..."
- Some of the brackets need to be made larger to fit their contents.

**Strengths And Weaknesses:**

Strengths:
- The paper is generally well written, and for the most part is easy to follow.
- The modified objective is a sensible and appealing alternative to the constrained maximum likelihood models (like those implemented in Mclust) which force certain equivalences in the covariance matrices of the different components. The penalty approach is both softer (and so does not explicitly enforce exact equality in component structure) and also is non-specific as it captures all types of "differences" between components. This objective alone is something which I believe will be interesting to the relevant community.
- The empirical performance of the clustering accuracy of the solutions when compared with direct ML and constrained ML solutions arising from EM and automatic differentiation shows promise in the method.

Weaknesses:
- I am afraid I cannot grasp the intuition behind MPKL as a model selection criterion, and I can't find a discussion in the paper for why this criterion makes sense.
- In relation to the above, I find many of the experiments to be fairly contrived to suit MPKL as a criterion. In particular, equal spherical covariance matrices directly lend themselves to being correctly identified by MPKL since, for example, any under-clustering will be picked up by the merged components having vastly different covariance from the individual components. The large cluster overlap scenario also seems to favour a criterion which rewards similar cluster means.
- The concept of a "spurious" solution seems to be pivotal, but the interpretation, in my experience, in the general literature is fairly vague. A precise definition of what the authors mean by a spurious solution would be helpful in interpreting their commentary. It is mentioned that spurious solutions resulting from covariance matrices of (at least nearly) non-full rank arise when the likelihood function is unbounded. If the likelihood objective is unbounded then how can we define a global maximum, and hence a ML solution?
- The theoretical contributions in the paper are, in my opinion, somewhat modest and this makes me question the appropriateness of TMLR as venue for publication.

---

> ### Comment · Reviewer_YoVD · 2022-08-25
> **Still not convinced**
>
> After much discussion with the authors, I unfortunately still do not fell particularly at ease with the method. I see the penalised objective used in SIA to be very sensible, and there is some evidence from the experiments that it works reasonably well. I do not understand the motivation for MPKL as a model selection criterion, and the explanation from the authors indicates it may only be a sensible criterion if the true covariances are similar (or am I misunderstanding), which is a very serious limitation, in my opinion. Based on the experimental results, MPKL seems at the very least reliable for selecting the values of w1 and w2 in SIA, but even that is questionable since all we see is the performance of the selected model. Maybe the other outputs from SIA were superior for clustering accuracy than the ones selected? Perhaps the authors can give some results on this. I don't see much evidence, though, that it can reliably be used to select the number of clusters, which is what the criterion seems to have been proposed for. According to the authors, you can use MPKL for SIA but must combine it with a likelihood based criterion; but you can use MPKL alone for other GMM based estimation methods?
>
> I am not sure I see a sensible penalty for the likelihood in GMM estimation which has reasonable performance in some experiments as being significant enough for publication in TMLR.

---

### Review · Reviewer_59ck · 2022-08-05

**Summary Of Contributions:**

In this paper, the authors present four different results in misspecified Gaussian mixture components (GMM). The authors assumed that the underlying clusters are single-class clusters and that the clustering would be used for classification. This is not clearly stated in the paper, but it is evident by the use of ARI as an error measure and the way they present their results.

The four results are:

* Section 3.1 shows that when the data is not Gaussian applying GMM learned by EM or GD can return clusters that do not represent the underlying classes. Of course, this is a problem that spectral clustering will excel at.

* Section 3.2 proves a theorem for an asymmetric coefficient (newly defined) for a mixture of 2 Gaussians when they are estimating a mixture of 3 Gaussians in 1d in which two of the Gaussians share the same mean.

* Section 4.2 the authors propose a new regularization to solve the GMM components, which is bounded, contrary to maximizing the likelihood.

* Section 4.4 the authors propose a new model selection criterion for the number of components.

**Broader Impact Concerns:**

There is no need for a broader impact section.

**Requested Changes:**

I think the paper should be rejected in its current form and a new submission created.

The paper's Intro and background are solid and attractive and show good prior work has been done. Section 3 should be deleted.

From Section 4, the authors should say that they are going to resolve the GMM model using a regularized log-likelihood maximization by adding a symmetric KL divergence for each pair in the mixture.

The experimental section should be remade by measuring the log-likelihood in a holdout set and using a different number of components and different datasets. Besides standard GMM with log-likelihood maximization, the authors should look for other algorithms that solve the same problem. I am not familiar with the literature, but I assume there must be paper regularizing the log-likelihood some other way.

Finally, the way that the symmetric KLD is minimized in the loss function does not make sense. I would expect to maximize the symmetric KLD between every pair because in that way we cover the most ground in the distribution. And ensures that the clusters are disjointed. Minimization of the KLD will force the clusters to be as much alike as possible, which for me is contra intuitive. This will need to be investigated further and why minimizing KL divergence is the best regularizer.

**Strengths And Weaknesses:**

[Weakness] This paper needs significant work before it can be converted into a publication. I think Section 3 is pointless and does not add anything to the known results. The simulation study just mentions that GMM can fail when the model is misspecified and the failures are measured in terms of the right classes, which should not be the measure of how good clustering is. If class labels are available, the problem is different. The theorem in Section 3.2 is for a particular problem with no interest and does not add any insights for the future.

[Strength] The algorithm in Section 4.2 and the theorem in Section 4.3 can be introduced directly after the background. This is probably the only interesting part of the paper. The authors propose to solve a regularized maximum likelihood problem, in which the regularizer is the symmetric KLD divergence between every component in the mixture model. This is explained in the paper in a very complicated way with a forward and a backward KL divergence. Adding this regularizer seems that makes the loss function bounded, which should kill the spurious solutions that can happen when one of the covariances gets very small. The problem in the paper is that the validity of this proposal is not well evaluated. Because the ARI is not the right way to evaluate GMMs.

[neural] Section 4.4 might have some value, but it should be a different paper and better introduced and compared. There is not enough material to evaluate this part of the paper.

---

### Decision · Action_Editors · 2022-09-13

**Recommendation:** Reject

**Comment:**

I would like to thank the authors for the extensive responses they provided. Unfortunately, the reviewers surfaced a number of concerns that remained after the discussion:

1. The findings reported in this work are of modest interest to the TMLR audience: the proposed setting (i.e., GMMs where the number of components corresponds to the number of subpopulation) is very restrictive when considering real data and hence of modest practical relevance to the ML community.
2. While claims in the submission were mostly clarified during the discussion, the characterisation of MPKL as a model selection criterion was misleading and its usefulness for model selection was not supported by clear evidence.

Reviewers provided a number of avenues to improve and clarify their work. I would encourage the authors to take them into account and provide better supporting evidence.